# CCRK/CDK20 regulates ciliary retrograde protein trafficking via interacting with BROMI/TBC1D32

Tatsuro Noguchi, Kentaro Nakamura¤, Yuuki Satoda, Yohei Katoh [ID]*,
Kazuhisa Nakayama [ID]*

Department of Physiological Chemistry, Graduate School of Pharmaceutical Sciences, Kyoto University,
Sakyo-ku, Kyoto, Japan

¤ Current address: Department of Physiological Chemistry, Graduate School of Pharmaceutical Sciences,
The University of Tokyo, Bunkyo-ku, Tokyo, Japan
* kazunaka@pharm.kyoto-u.ac.jp (KN); ykatoh@pharm.kyoto-u.ac.jp (YK)

pone.0258497

IRELAND

**Data Availability Statement:** All relevant data are
within the manuscript and its Supporting
Information files.

## Abstract

CCRK/CDK20 was reported to interact with BROMI/TBC1D32 and regulate ciliary Hedge-
hog signaling. In various organisms, mutations in the orthologs of CCRK and those of the
kinase ICK/CILK1, which is phosphorylated by CCRK, are known to result in cilia elongation.
Furthermore, we recently showed that ICK regulates retrograde ciliary protein trafficking
and/or the turnaround event at the ciliary tips, and that its mutations result in the elimination
of intraflagellar transport (IFT) proteins that have overaccumulated at the bulged ciliary tips
as extracellular vesicles, in addition to cilia elongation. However, how these proteins cooper-
ate to regulate ciliary protein trafficking has remained unclear. We here show that the pheno-
types of *CCRK*-knockout (KO) cells closely resemble those of *ICK*-KO cells; namely, the
overaccumulation of IFT proteins at the bulged ciliary tips, which appear to be eliminated as
extracellular vesicles, and the enrichment of GPR161 and Smoothened on the ciliary mem-
brane. The abnormal phenotypes of *CCRK*-KO cells were rescued by the exogenous
expression of wild-type CCRK but not its kinase-dead mutant or a mutant defective in
BROMI binding. These results together indicate that CCRK regulates the turnaround pro-
cess at the ciliary tips in concert with BROMI and probably via activating ICK.

## Introduction

The crucial roles of primary cilia in embryonic development and adult tissue homeostasis are
widely acknowledged by the fact that ciliary dysfunctions cause genetically heterogeneous dis-
orders with a wide spectrum of clinical manifestations, collectively referred to as the ciliopa-
thies [1,2]. Cilia are axonemal microtubule-based protrusions surrounded by the ciliary
membrane, which are found in a variety of eukaryotic cells. Via specific receptors and ion
channels on the ciliary membrane, cilia function as cellular antennae for extracellular stimuli,
such as fluid flow, and for extracellular signaling molecules, such as the Hedgehog (Hh) family
of morphogens [3,4].

**Funding:** This work was supported in part by grants from the Japan Society for the Promotion of Science (JSPS) (grant numbers 19H00980 and 20H04904 to K. Nakayama, and 18H02403 and 21H02427 to Y. Katoh); and a grant of JRPs-LEAD with UKRI from JSPS (grant number JPJSJRP20181701 to K. Nakayama).

**Competing interests:** No authors have competing interests.

**Abbreviations:** BBS, Bardet-Biedl syndrome; CCRK, cell cycle-related kinase; CILK1, ciliogenesis-associated kinase 1; ECV, extracellular vesicle; GST, glutathione *S*-transferase; hTERT-RPE1, human telomerase reverse transcriptase-immortalized retinal pigment epithelial 1; ICK, intestinal cell kinase; IFT, intraflagellar transport; KO, knockout; MAK, male germ cell-associated kinase; mChe, mCherry; MEF, mouse embryonic fibroblast; Nb, nanobody; PP, phosphoprotein phosphatase; SAG, Smoothened Agonist; SMO, Smoothened; VIP, visible immunoprecipitation; WT, wild-type.

To achieve these specialized functions, the protein compositions of the ciliary membrane and cilioplasm are maintained to be distinct from those of the plasma membrane and cytoplasm, respectively, by the presence of the ciliary gate comprising transition fibers and the transition zone as a permeability/diffusion barrier [5]. Not only bidirectional protein trafficking along the axonemal microtubules, but also import and export of ciliary proteins across the ciliary gate are mediated by the intraflagellar transport (IFT) machinery, which is often referred to as IFT trains or IFT particles [6–8]. IFT particles were originally purified from a flagellar fraction of the green alga *Chlamydomonas* [9,10]. Two multisubunit complexes, IFT-A and IFT-B, are the main components of the IFT machinery [11,12]. The IFT-B complex contains 16 subunits and mediates anterograde (ciliary base-to-tip direction) trafficking of proteins driven by heterotrimeric kinesin-II, and the IFT-A complex comprises six subunits and mediates retrograde trafficking driven by the dynein-2 motor complex [12,13]. Besides these roles, the IFT-B complex participates in the export of membrane proteins across the ciliary gate in conjunction with the BBSome, which is composed of eight proteins encoded by the causative genes of Bardet-Biedl syndrome (BBS) [14–18]. On the other hand, the IFT-A complex mediates the import of membrane proteins with the aid of the TULP3 adaptor protein [19–22].

In addition to the bidirectional trafficking along the axoneme, regulation of the turnaround of the IFT machinery at the ciliary tip is also crucial for ciliary protein trafficking [11]. Live imaging analyses of IFT particles in flagella/cilia of *Chlamydomonas*, *Caenorhabditis elegans*, and trypanosome indicated that disassembly of the particles and mixing of the IFT-A and IFT-B components occur at the ciliary tip [23–25]. On the other hand, cryoelectron tomographic studies of *Chlamydomonas* IFT trains indicated the distinct configurations of the anterograde and retrograde trains [26,27]. Furthermore, the change in IFT direction at the tip entails switching of the anterograde kinesin motor to the retrograde dynein motor. In line with this motor switching, the structure of the human dynein-2 complex revealed by cryoelectron microscopy suggested that dynein-2 is transported as an inactive cargo of the anterograde trains [28].

Candidates of crucial regulators of the turnaround event are intestinal cell kinase (ICK; recently renamed as CILK1 for ciliogenesis-associated kinase 1) and its sperm- and retina-specific paralog, male germ cell-associated kinase (MAK), both of which belong to the MAP kinase superfamily [29] and are localized at the ciliary tips [30–33]. Mutations in the *ICK* gene in humans cause ciliopathies [34–36], and knockout (KO) and mutant *Ick* mice manifest ciliopathy phenotypes [32,37,38]. Previous studies proposed that the phosphorylation of KIF3A, a kinesin-II subunit, by ICK is crucial for the regulation of ciliary protein trafficking [32,39]; however, a recent study suggested that KIF3A phosphorylation is dispensable for ciliary function [40].

ICK and MAK are phosphorylated at the canonical TDY motif by another kinase, cell cycle-related kinase (CCRK, also known as CDK20) [41,42], and phosphorylation of an ICK ortholog in *Chlamydomonas* is not detected in a mutant strain of a CCRK ortholog [43]. In agreement with the regulation of ICK and MAK by CCRK, mutations in *CCRK* and its interacting partner *BROMI* (also known as TBC1D32) in humans cause ciliopathies, and their mutant/KO mice manifest defective embryonic development caused by dysregulated Hh signaling [44–47]. Moreover, mutation/knockdown/KO of not only ICK and MAK, but also CCRK, as well as their homologs in a variety of organisms are known to result in a long cilia/flagella phenotype [31,37,43,48–56].

On the other hand, we recently showed that not only is the average ciliary length longer but also the variation in ciliary length is larger in *ICK*-KO cells than in control cells, and that excessively accumulated proteins at the bulged ciliary tips of *ICK*-KO cells are eliminated as

extracellular vesicles (ECVs) [30]. A broader distribution of ciliary/flagellar lengths as well as the accumulation of ciliary/flagellar proteins at the distal tips was previously reported in cells derived from *Ccrk*-deficient mice and in a *Chlamydomonas* null mutant of its CCRK homolog [44,57]. Thus, the phenotypes of *CCRK*-deficient cells closely resemble those of *ICK*-KO cells. However, studies on CCRK and its orthologs to date have focused mainly on ciliary length, and not on the regulation of ciliary protein trafficking by CCRK, which underlies the control of ciliary length and ECV release.

In this study, we characterized the interaction of CCRK/CDK20 with BROMI/TBC1D32. We then established *CCRK*-KO cells and compared the phenotypes of *CCRK*-KO cells with those of *ICK*-KO cells. We also analyzed the phenotypes of *CCRK*-KO cells exogenously expressing various CCRK constructs. Our results showed that *CCRK*-KO cells accumulated IFT proteins at their bulged ciliary tips, which appeared to be eliminated as ECVs. The abnormal phenotypes of *CCRK*-KO cells were rescued by the exogenous expression of wild-type (WT) CCRK but not its kinase-dead mutant or a mutant defective in BROMI binding. These results altogether indicate that CCRK regulates the IFT turnaround process in concert with BROMI, probably via activating ICK.

## Materials and methods

### Plasmids, antibodies, and reagents

cDNAs for human CCRK (NM_001039803.3) and BROMI (NM_001367760.1) were obtained from a cDNA library by PCR amplification. The human ICK cDNA was kindly provided by Takahisa Furukawa (Osaka University) [32]. Expression vectors for CCRK and BROMI, and their deletion and point mutants, and vectors for the production of lentiviruses expressing CCRK and ICK, as listed in S1 Table, were constructed. Antibodies used in this study are listed in S2 Table. Glutathione *S*-transferase (GST)-tagged anti-GFP Nbs prebound to glutathione–Sepharose 4B beads were prepared as described previously [58,59]. Smoothened Agonist (SAG) was purchased from Enzo Life Sciences.

**VIP assay and immunoblotting analysis.** The visible immunoprecipitation (VIP) assay and subsequent immunoblotting analysis were performed as described previously [59,60]; experimental details of the VIP assay were described in detail elsewhere [58]. Briefly, HEK293T cells (RBC2202; RIKEN BioResource Research Center) grown on a 6-well plate were cotransfected with expression vectors for EGFP-fused and mChe-fused proteins using Polyethylenimine Max (Polysciences), and cultured for 24 h in high glucose Dulbecco's modified Eagle's medium (DMEM) (Nacalai Tesque) supplemented with 5% fetal bovine serum (FBS). The transfected cells were then lysed in 250 μL of HMDEKN cell-lysis buffer (10 mM HEPES [pH 7.4], 5 mM $MgSO_4$, 1 mM DTT, 0.5 mM EDTA, 25 mM KCl, and 0.05% NP-40) containing protease inhibitor cocktail (Nacalai Tesque) by incubation for 15 min on ice. The lysates were then centrifuged at $16,100 \times g$ for 15 min, and supernatants (200 μL) were transferred to a 0.2 mL 8-tube strip, to which GST-tagged anti-GFP Nbs prebound to glutathione–Sepharose 4B beads (approximately 5 μL bed volume of the beads; GE Healthcare) was added, and incubated for 1 h at 4°C with constant rotation of the tube strip. After brief centrifugation at $2,000 \times g$ for 10 sec, the precipitated beads were washed three times with lysis buffer (180 μL), transferred to a 96-well glass-bottom plate (AGC Techno Glass), and observed under a BZ-8000 all-in-one type microscope (Keyence) with a 20×/0.75 NA objective lens under fixed conditions (for green fluorescence: sensitivity ISO 400, exposure 1/10 s; and for red fluorescence: sensitivity ISO 800, exposure 1/5 s).

The beads bearing fluorescent proteins were then subjected to immunoblotting analysis. Proteins on the beads were separated by SDS-polyacrylamide gel electrophoresis and electroblotted

onto an Immobilon-P membrane (Merck Millipore). The membrane was blocked in 5% skimmed milk and incubated sequentially with primary antibody and peroxidase-conjugated secondary antibody. Protein bands were detected using a Chemi-Lumi One L kit (Nacalai Tesque).

**Establishment of *CCRK*-KO cell lines.** The strategy for the disruption of genes in human telomerase reverse transcriptase-immortalized retinal pigment epithelial 1 (hTERT-RPE1) cells (CRL-4000, American Type Culture Collection) by the CRISPR/Cas9 system using homology-independent DNA repair (version 2 method) was performed as previously described in detail [61], with slight modifications [62–64]. In brief, single-guide RNA (sgRNA) sequences targeting the human *CCRK/CDK20* gene (see S3 Table) were designed using CRISPOR [65]. Double-stranded oligonucleotides for the target sequences were inserted into the all-in-one sgRNA expression vector pHiFiCas9-2×sgRNA (Addgene 162277) [66], in which the eSpCAS9 sequence in peSpCAS9(1.1)-2×sgRNA [61] was replaced with the high fidelity Cas9 sequence, HiFiCas9 [67]. hTERT-RPE1 cells grown on a 12-well plate to approximately $3.0 \times 10^5$ cells were transfected with the all-in-one vector and the donor knock-in vector, pDonor-tBFP-NLS-Neo(universal) (Addgene 80767), using X-tremeGENE9 Reagent (Roche Applied Science). After selection of the transfected cells in the presence of G418 (600 μg/mL), cells were isolated using an SH800 Series cell sorter (SONY) at the Medical Research Support Center, Graduate School of Medicine, Kyoto University. Genomic DNA was extracted from the isolated cells, and analyzed by PCR using GoTaq Master Mixes (Promega) and three sets of primers (S3 Table) to distinguish the following three states of integration of the donor knock-in vector: forward integration, reverse integration, and no integration with a small indel (for example, see [19]). The disruption of both *CCRK* alleles was confirmed by direct sequencing of the PCR products.

The *ICK*-KO cell line #ICK-4-6 was established as described previously [30].

**Preparation of cells stably expressing EGFP-fused CCRK and mChe-fused ICK constructs.** Lentiviral vectors for the expression of various CCRK constructs were prepared by a previously described method [68]. Briefly, pRRLsinPPT-EGFP-N-CCRK or a vector encoding its mutant construct was transfected into HEK293T cells along with the packaging plasmids (pRSV-REV, pMD2.g, and pMDLg/pRRE; kind gifts from Peter McPherson, McGill University; [69]). Culture media were replaced 8 h after transfection, and those containing viral particles were collected at 24, 36, and 48 h after transfection. The collected media were passed through a 0.45-μm filter and centrifuged at $32,000 \times g$ at 4˚C for 4 h. The precipitates containing lentiviral particles were resuspended in Opti-MEM (Invitrogen). The preparation of lentiviral vectors for the ICK constructs was as described previously [30]. The lentiviral suspension was added to the culture medium of hTERT-RPE1 cells or *CCRK*-KO cell lines established as described above. After a 24-h incubation, the cells were used for subsequent analyses.

**Immunofluorescence analysis and live cell imaging.** hTERT-RPE1, *CCRK*-KO, and *ICK*-KO cells were cultured in DMEM/F-12 (Nacalai Tesque) supplemented with 10% FBS and 0.348% sodium bicarbonate. To induce ciliogenesis, cells were grown on coverslips to 100% confluence, and starved for 24 h in Opti-MEM containing 0.2% bovine serum albumin.

Unless otherwise noted, immunofluorescence analysis was performed as described previously [30,64,70]. Cells on coverslips were fixed with 3% paraformaldehyde for 5 min at 37˚C, permeabilized with 100% methanol for 5 min at −20˚C (for experiments shown in Fig 4), or fixed and permeabilized with 100% methanol for 5 min at −20˚C (for experiments shown in Figs 2, 3, 5 and 6), and washed three times with phosphate-buffered saline. The fixed/permeabilized cells were blocked with 10% FBS, incubated sequentially with primary and secondary antibodies diluted in Can Get Signal Immunostain Solution A (Toyobo) (for the detection of SMO) or in 5% FBS (for the detection of the other proteins), and observed using an Axio Observer microscope (Carl Zeiss). For data shown in Fig 5J, cells were fixed with 3% paraformaldehyde for 15 min at room temperature, permeabilized with 100% methanol for 5 min at

−20˚C, washed three times with phosphate-buffered saline. After sequential incubation with primary and secondary antibodies, the cells were observed using a confocal laser scanning microscope (Nikon, AX-R). A region of interest (ROI) was created by drawing a line of 3-point width along the signal of ARL13B or Ac-tubulin within cilia using a segmented line tool in the ZEN 3.1 imaging software (Carl Zeiss). For the correction of local background intensity, the ROI was duplicated and set to a nearby region. Statistical analyses were performed using GraphPad Prism8 (Version 8.4.3; GraphPad Software, Inc.).

Live-cell imaging to observe the release of ECVs from cilia was performed as described previously [30,71,72]. Briefly, *CCRK*-KO cells expressing EGFP-fused ARL13B(ΔGD) were serum-starved for 24 h on a glass-bottom culture dish (MatTek) and observed under an A1R-MP confocal laser-scanning microscope (Nikon). Time-lapse images were acquired sequentially every 5 min and analyzed using NIS-Elements imaging software (Nikon).

## Results

### Interaction of CCRK with BROMI

Although CCRK/CDK20 was shown to interact with BROMI/TBC1D32 approximately 10 years previously [45], the interaction has not been characterized in detail since then. Therefore, we first characterized the CCRK–BROMI interaction using the visible immunoprecipitation (VIP) assay, followed by conventional immunoblotting analysis; the VIP assay is a convenient and versatile method that enables visual detection of protein–protein interactions [59,73].

Lysates of HEK293T cells coexpressing mCherry (mChe)-fused BROMI and one of the various CCRK constructs fused to EGFP were processed for immunoprecipitation with glutathione *S*-transferase (GST)-tagged anti-GFP nanobody (Nb) prebound to glutathione–Sepharose beads. Red signals were observed on the precipitated beads from the lysates of cells coexpressing EGFP-fused CCRK(WT) and BROMI-mChe, indicating their interaction (Fig 1E, column 2). When the CCRK protein was divided into the CDK-like kinase domain (residues 1–289; see Fig 1A) and the C-terminal noncatalytic region (residues 290–346), the latter was found to interact with BROMI-mChe (Fig 1E, compare columns 4 and 5). In line with this, a kinase-dead CCRK mutant (K33R) retained the ability to interact with BROMI-mChe (Fig 1E, column 3). When CCRK was truncated from the C-terminus, truncation of the last 16-amino acids (the CCRK(1–330) construct) was sufficient to abolish its interaction with BROMI (column 6). The VIP results were confirmed by immunoblotting analysis. As shown in Fig 1F, EGFP-fused CCRK(WT), CCRK(290–346), and CCRK(K33R) coimmunoprecipitated BROMI-mChe (lanes 2, 5, and 3), whereas CCRK(1–289) and CCRK(1–330) did not (lanes 4 and 6). These results indicate that the C-terminal noncatalytic region of CCRK is responsible for its interaction with BROMI. The noncatalytic region is moderately conserved in vertebrates (Fig 1B), but not conserved in non-vertebrate species, and is predicted to be relatively disordered in the AlphaFold Protein Structure Database [74].

BROMI/TBC1D32 has a Tre-2/Bub2/Cdc16 (TBC) domain at its C-terminus. The TBC domain is evolutionarily conserved among GTPase-activating proteins (GAPs) for RAB GTPases [75], although BROMI was suggested not to act as a RAB-GAP, as it lacks the Arg and Gln residues within the catalytic motifs that are essential for GAP activity [45]. When the BROMI protein was divided into the C-terminal TBC domain (residues 1,102–1,298) and the residual N-terminal region (residues 1–1,101), neither regions retained the ability to interact with CCRK (Fig 1G and 1H; compare column/lane 2 with columns/lanes 3 and 4). When truncated from the N-terminus, the BROMI(157–1,298) and BROMI(182–1,298) constructs retained to interact with CCRK, but BROMI(243–1,298) had a reduced ability (Fig 1G and 1H, columns/lanes 5–7). On the other hand, the C-terminal truncation mutant BROMI(1–1,190)

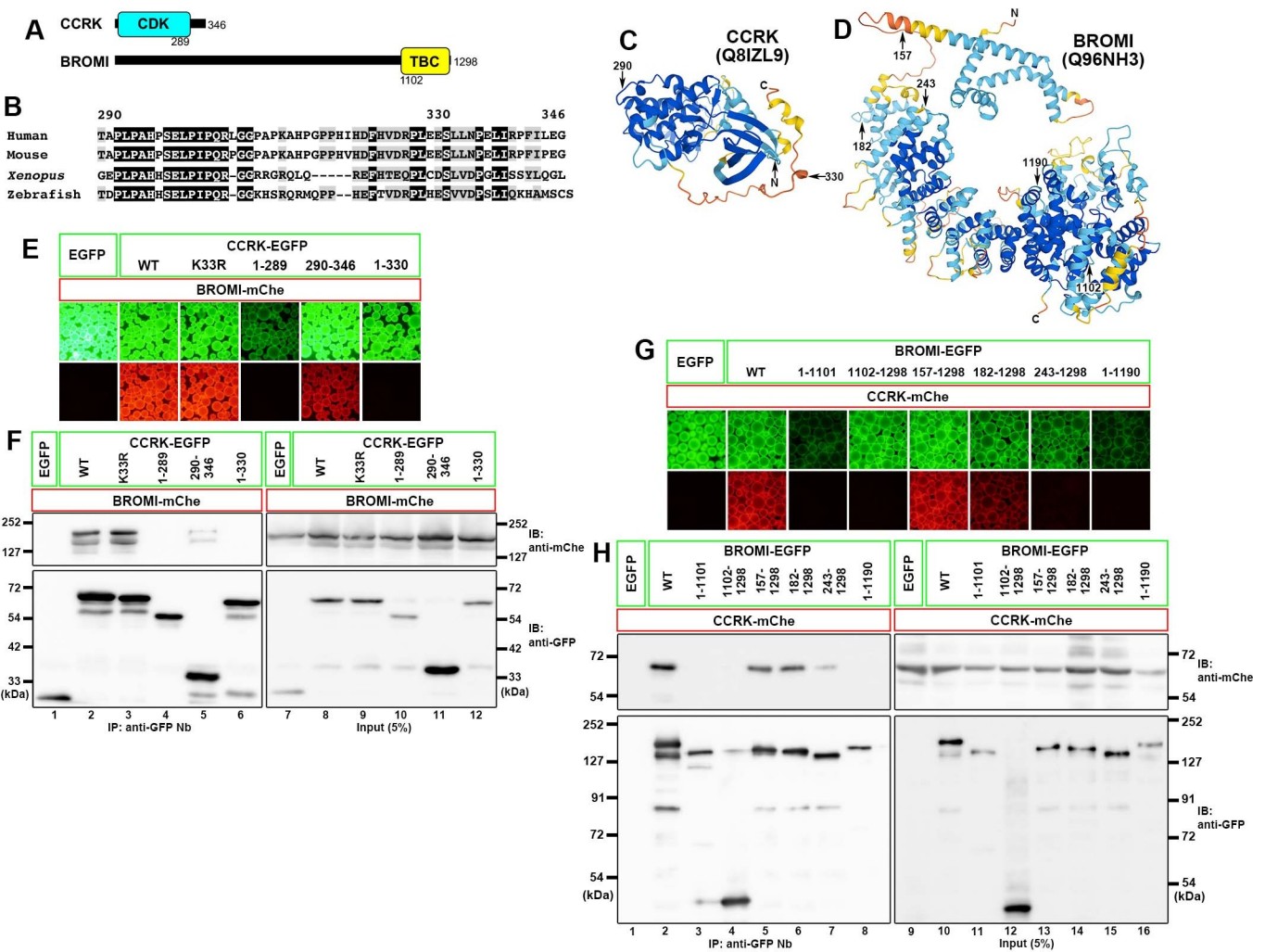

**Fig 1. Interaction of CCRK with BROMI.** (A) Schematic representation of the domain organizations of CCRK and BROMI. (B) Sequence alignment of the C-terminal noncatalytic region of vertebrate CCRK. Amino acid residues identical among all members are shown in a black background, and those with conservative substitutions are shown in a grey background. (C, D) The predicted 3D structures of human CCRK (C) and BROMI (D) in the AlphaFold Protein Structure Database [74]. The positions of important residues in the CCRK and BROMI constructs are shown. (E, F) Interaction of the C-terminal noncatalytic region of CCRK with BROMI. Lysates of HEK293T cells coexpressing the EGFP-fused CCRK constructs as indicated and BROMI-mChe were processed for the VIP assay using GST-fused anti-GFP Nb (E) followed by immunoblotting analysis using an anti-mChe or anti-GFP antibody (F). (G, H) Requirement of both the N-terminal and C-terminal regions of BROMI for its interaction with CCRK. Lysates of HEK293T cells coexpressing EGFP-fused BROMI constructs as indicated and CCRK-mChe were processed for the VIP assay (G) followed by immunoblotting analysis (H), as described above.

lacked CCRK-interacting ability (column/lane 8). Thus, in the BROMI protein, the C-terminal region is essential for and the N-terminal region also contributes to its interaction with CCRK. As the C-terminal region forms a rigid structure, and residue 243, but not residue 157 or 182, appears to be included in the ordered structure in the predicted 3D structure of BROMI (Fig 1D) in the AlphaFold Protein Structure Database [74], the overall ordered structure may be disrupted in the BROMI(243–1,298) and BROMI(1–1,190) constructs.

## Phenotypes of *CCRK*-KO cells resemble those of *ICK*-KO cells

By applying the modified CRISPR/Cas9 system [61] to hTERT-RPE1 cells, we established *CCRK*-KO cell lines and characterized their phenotypes. Two independent KO cell lines

(#CCRK-1-1 and #CCRK-2-4), which were established using distinct target sequences (S1 Fig), were selected for the following analyses. When immunostained for ARL13B (a marker for the ciliary membrane) and acetylated α-tubulin (Ac-tubulin; a marker for axonemal microtubules), the cilia of both the *CCRK*-KO cell lines as well as the *ICK*-KO cell line, which we previously established [30], were found to be significantly longer than those of control RPE1 cells (Fig 2A–2D; also see Fig 2M). The long cilia often appeared to be bending or winding, indicative of their growth on the ventral surface of the cells [30,76]. The long cilia phenotype is consistent with previous observations of mutants of CCRK and ICK homologs in a variety of organisms [31,37,43,44,48–56]. In addition to the greater average ciliary lengths, the variations in ciliary length appeared to be greater in *CCRK*-KO and *ICK*-KO cells than in control cells (Fig 2M), which is consistent with a previous study showing a broader distribution of ciliary lengths in *Ccrk* null MEFs than in control MEFs [44]. The similarity in the ciliary length distribution between *CCRK*-KO and *ICK*-KO cells suggests the release of ECVs from the ciliary tip of *CCRK*-KO cells as in *ICK*-KO cells [30] (see below).

We then compared the localizations of the IFT-B and IFT-A proteins between control RPE1 cells and *CCRK*-KO and *ICK*-KO cells. In control RPE1 cells, IFT88 (an IFT-B subunit) was detected mainly at the base of cilia, with a small proportion detected at the tip (Fig 2E). By contrast, the proportion of IFT88 found at the ciliary tip was significantly increased in *CCRK*-KO cells (Fig 2F and 2G) as well as in *ICK*-KO cells (Fig 2H; also see Fig 2N). The accumulation of IFT88 at the ciliary tip in *CCRK*-KO RPE1 cells was essentially the same as that observed in MEFs from *Ccrk*-KO mice [44]. Similar results were obtained for the localization of IFT140 (an IFT-A subunit) (Fig 2I–2L; also see Fig 2O).

We previously showed that, in *ICK*-KO cells and *IFT144*-KO cells, as well as in cells with a compromised interaction between the IFT-A and IFT-B complexes, ciliary proteins that are excessively accumulated at the ciliary tips owing to an impairment in the turnaround process are packaged into ECVs and eliminated to relieve cilia of the stress of protein overaccumulation [30,71]. This was also the case in *CCRK*-KO cells. When cells were immunostained for IFT88, ARL13B, and FOP (a basal body marker; also known as CEP43), we often detected the presence of punctate structures, which are positive for both IFT88 and ARL13B but negative for FOP, in images of *CCRK*-KO cells (Fig 2Q, d–f) as well as in those of *ICK*-KO cells (Fig 2R, d–f); such punctate structures were not observed in control RPE1 cells (Fig 2P). As shown in S1 and S2 Videos and in Fig 2S and 2T, the release of ECVs from ciliary tips was often observed by live-cell imaging of *CCRK*-KO cells expressing ARL13B(ΔGD)-EGFP. These observations, in conjunction with our previous studies [30,71], indicate that the IFT turnaround process at the tip is impaired in the absence of CCRK, and indicate that overaccumulated proteins at the ciliary tips are packaged within ECVs and eliminated to relieve cilia from excessive stress. As bulbous ciliary tips were observed in *Ccrk*-KO mice [44,77], it is likely that ECV release also occurs *in vivo* in the absence of CCRK.

The change in direction of IFT at the distal tips entails switching of the anterograde kinesin motor to the retrograde dynein motor [11]. We therefore analyzed the localization of EGFP-fused KAP3 (a nonmotor subunit of heterotrimeric kinesin-II; also known as KIFAP3) and DYNC2LI1 (a subunit of the dynein-2 complex) that were stably expressed in *CCRK*-KO and *ICK*-KO cells. In control RPE1 cells, KAP3-EGFP was faintly detectable at the ciliary base (Fig 3A). In striking contrast, distinct signals of KAP3-EGFP were found at the distal tips as well as at the base in *CCRK*-KO and *ICK*-KO cells (Fig 3B and 3C; also see Fig 3G). These observations are in line with previous studies showing that in *C. elegans* mutants of both CCRK and ICK homologs, kinesin-II inappropriately enters the distal segments of sensory cilia [50,54], although it should be noted that two types of kinesin-2 participate in intraciliary trafficking in *C. elegans* [78]. Similar results were obtained for EGFP-DYNC2LI1, which was faintly detected

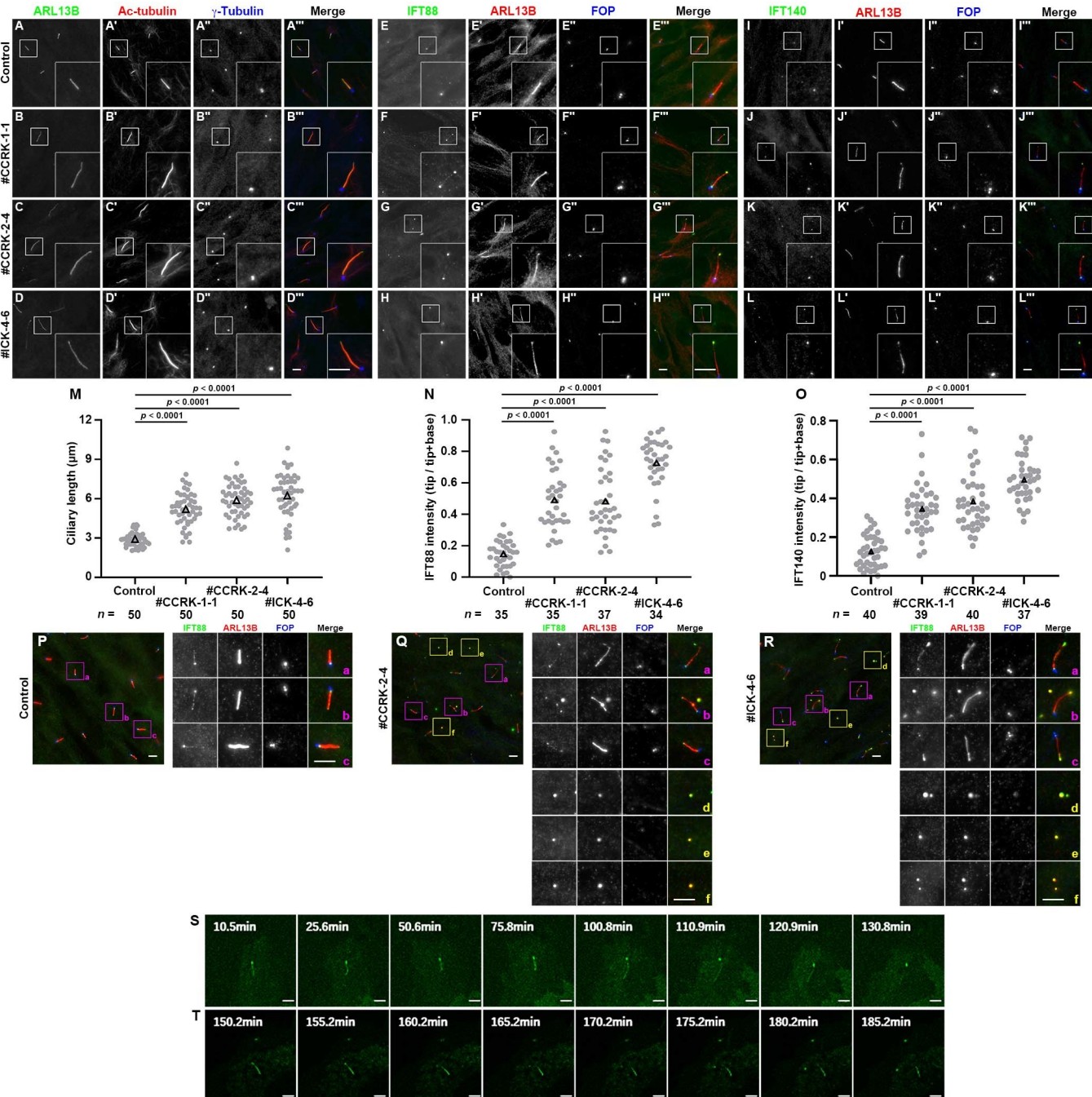

**Fig 2. Cilia elongation and the accumulation of IFT proteins at the ciliary tips of *CCRK*-KO cells.** (A–L) Control RPE1 cells (A, E, I), the *CCRK*-KO cell lines #CCRK-1-1 (B, F, J) and #CCRK-2-4 (C, G, K), and the *ICK*-KO cell line #ICK-4-6 (D, H, L) were serum-starved for 24 h to induce ciliogenesis, and triple immunostained for ARL13B, Ac-tubulin, and γ-tubulin (A–D), IFT88, ARL13B, and FOP (E–H), or IFT140, ARL13B, and FOP (I–L). Scale bars, 5 μm. (M) Ciliary lengths of the cells shown in (A)–(D) were measured and expressed as scatter plots. (N, O) Relative staining intensities of IFT88 and IFT140 at the ciliary tips and base in the cells shown in (E)–(H) and (I)–(L), respectively, were estimated, and the intensity ratios of tip/(tip + base) were expressed as scatter plots. The triangles indicate the means. Statistical significances were calculated using one-way ANOVA followed by the Dunnett's multiple comparison test. (P–R) Control RPE1 cells (P), the *CCRK*-KO cell line #CCRK-2-4 (Q), and the *ICK*-KO cell line #ICK-4-6 (R) were triple immunostained for IFT88, ARL13B, and FOP as described above. The boxed regions labeled a–f were 2.5-fold magnified and shown at the right. (S, T) The release of ECVs from the ciliary tips of the *CCRK*-KO cell line #CCRK-2-4 stably expressing ARL13B(ΔGD)-EGFP. Time-lapse images of S1 and S2 Videos are shown. Scale bars, 5 μm.

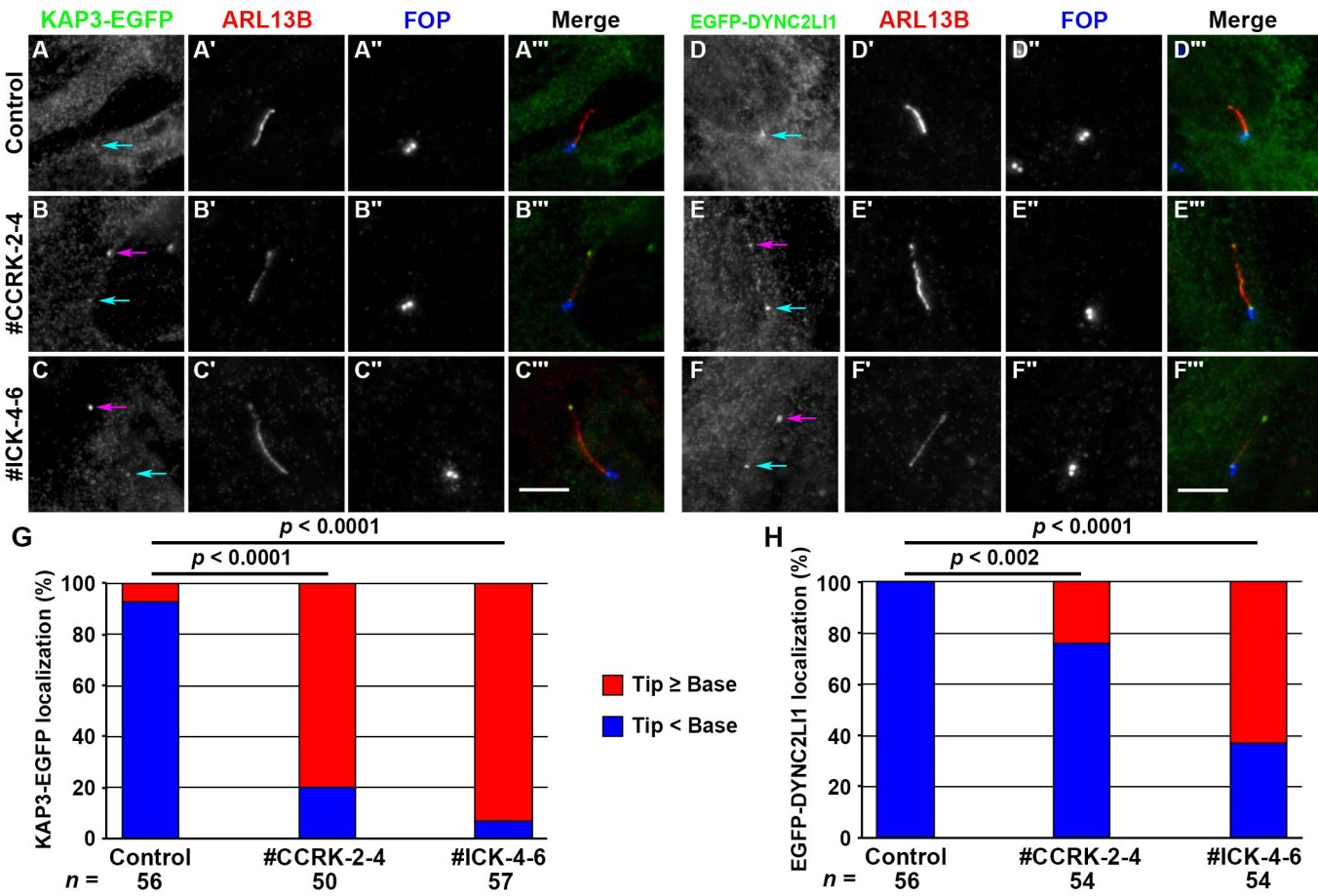

**Fig 3. Accumulation of kinesin-II and dynein-2 at the ciliary tips of *CCRK*-KO cells.** Control RPE1 cells (A, D), the *CCRK*-KO cell line #CCRK-2-4 (B, E), and the *ICK*-KO cell line #ICK-4-6 (C, F) stably expressing KAP3-EGFP (A–C) or EGFP-DYNC2LI1 (D–F), were serum-starved for 24 h, and triple immunostained for GFP, ARL13B, and FOP. Cyan and magenta arrows indicate the positions of KAP3-EGFP and EGFP-DYNC2LI1 signals at the ciliary base and tip, respectively. Scale bars, 5 μm. (G, H) The localization of KAP3-EGFP (G) and EGFP-DYNC2LI1 (H) in individual cells was classified as 'mainly ciliary tip (Tip ≥ Base)' and 'mainly ciliary base (Tip < Base)', and the number of cells in each category was counted. The percentages of these categories are expressed as stacked bar graphs. Statistical significances were calculated using one-way ANOVA followed by the Dunnett's multiple comparison test. Note that, owing to the relatively high cytoplasmic background staining for KAP3-EGFP and EGFP-DYNC2LI1, we could not precisely estimate the staining intensities.

at the ciliary base in control cells, whereas it was substantially enriched at the ciliary tip in *CCRK*-KO and *ICK*-KO cells (Fig 3D–3F; also see Fig 3H); note that ciliary tip accumulation of DYNC2LI1 was less prominent in *CCRK*-KO cells than in *ICK*-KO cells. These observations suggest two possibilities, although these are not mutually exclusive. One is that motor switching at the distal tips was compromised, at least partially, in the absence of CCRK or ICK, and the other is that, after motor switching, both the kinesin and the dynein motors were accumulated at the tips owing to the impaired turnaround process in the absence of CCRK or ICK (see Discussion).

We then compared the localizations of GPR161 and Smoothened (SMO) in control RPE1, *CCRK*-KO, and *ICK*-KO cells. GPR161 and SMO are ciliary GPCRs belonging to class A and class F, respectively, and are involved in the negative and positive regulation of Hh signaling, respectively [4,79]. In control RPE1 cells, GPR161 is found within cilia (Fig 4A), whereas SMO is barely detectable within cilia (Fig 4I) under basal conditions (–SAG). When the cells are treated with Smoothened Agonist (SAG), a small molecule that activates the Hh pathway via binding SMO, GPR161 exits cilia (Fig 4E) whereas SMO enters cilia (Fig 4M). In contrast, the

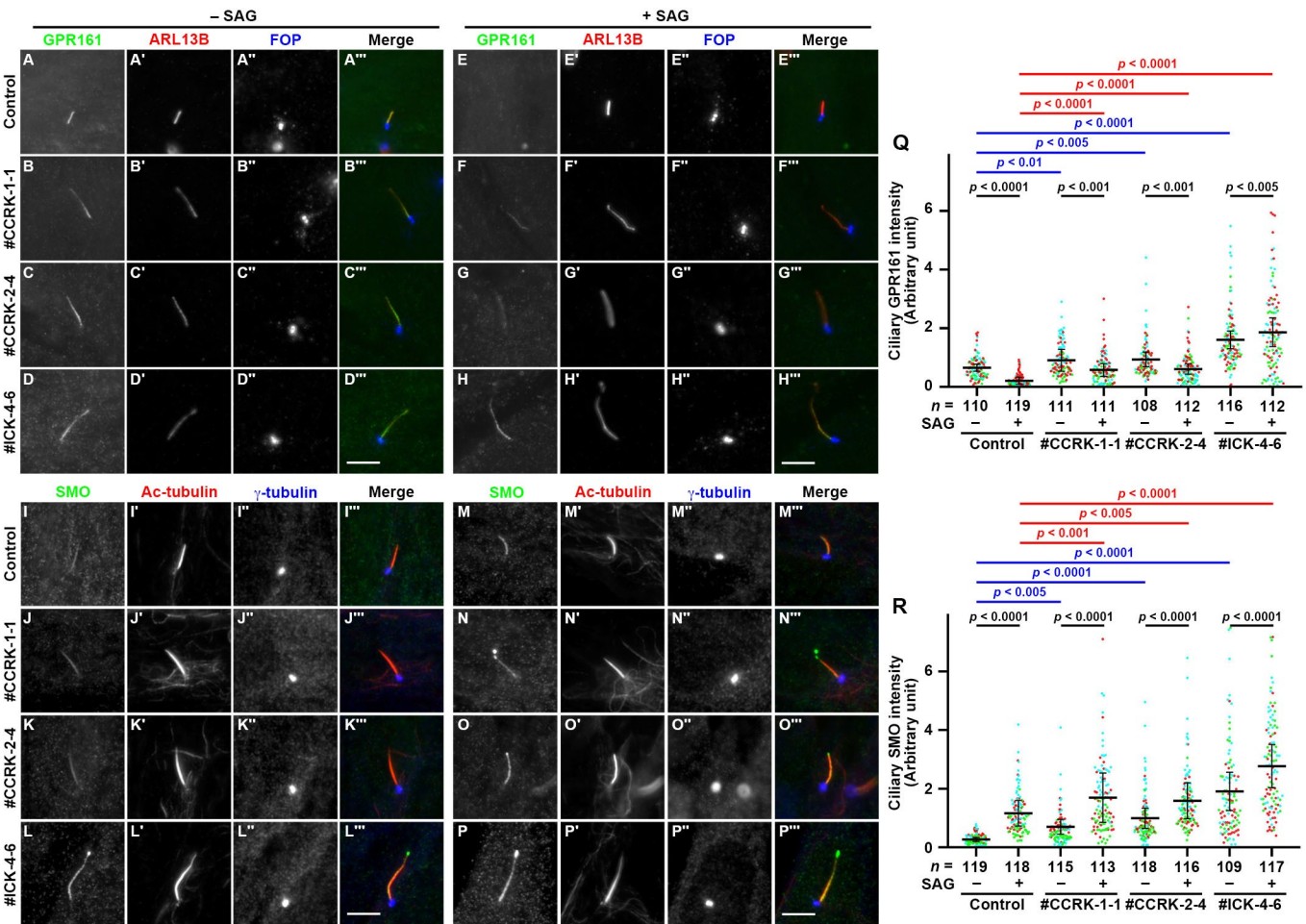

**Fig 4. Accumulation of GPR161 and SMO within cilia in *CCRK*-KO cells.** (A–P) Control RPE1 cells (A, E, I, M), the *CCRK*-KO cell lines #CCRK-1-1 (B, F, J, N) and #CCRK-2-4 (C, G, K, O), and the *ICK*-KO cell line #ICK-4-6 (D, H, L, P) were serum-starved for 24 h, and further incubated for 24 h without (−SAG; A–D, I–L) or with (+SAG; E–H, M–P) 200 nM SAG. The cells were triple immunostained for GPR161 (A–H), ARL13B (A′–H′), and FOP (A″–H″), or for SMO (I–P), Ac-tubulin (I′–P′), and γ-tubulin (I″–P″). Scale bars, 5 μm. (Q, R) The relative ciliary staining intensities of GPR161 and SMO shown in (A–H) and (I–P), respectively, were estimated and expressed as scatter plots. The total number of cells analyzed (*n*) are indicated. In the scatter plots, different colored dots represent three independent experiments, horizontal lines indicate the means, and error bars are the SDs. Statistical significances among multiple cell lines were calculated using one-way ANOVA followed by the Tukey multiple comparison test, and those between two groups (−SAG and +SAG) were calculated using the Bonferroni-adjusted Student *t*-test.

basal ciliary level of GPR161 was significantly higher in *CCRK*-KO and *ICK*-KO cells than in control cells, with a more robust increase in *ICK*-KO cells (Fig 4A–4D; also see Fig 4Q). Under SAG-stimulated conditions, the export of ciliary GPR161 was significantly suppressed in *CCRK*-KO and *ICK*-KO cells, although the suppression was less robust in *CCRK*-KO cells (Fig 4E–4H; also see Fig 4Q). On the other hand, under basal conditions, SMO levels were significantly high in *CCRK*-KO and *ICK*-KO cilia (Fig 4J–4L; also see Fig 4R). Upon stimulation with SAG, SMO gained entry into cilia in both *CCRK*-KO and *ICK*-KO cells, as in control cells (Fig 4N–4P; also see Fig 4R); it is notable that SMO was often found in bulbous tip structures of SAG-stimulated *CCRK*-KO and *ICK*-KO cells (for example, see Fig 4N and 4P), indicative of subsequent ECV release. Thus, ciliary levels of GPR161 and SMO were higher in *CCRK*-KO and *ICK*-KO cells than in control cells, suggesting that retrograde trafficking and/or exit from cilia of these GPCRs were compromised in the absence of CCRK, as in the absence of ICK.

## Normal function of CCRK requires its interaction with BROMI

To rule out the possibility that the abnormal phenotypes of *CCRK*-KO cells resulted from off-target effects of the CRISPR/Cas9 system, we then analyzed whether the phenotype was rescued by the exogenous expression of CCRK. Exogenously expressed CCRK(WT)-EGFP in *CCRK*-KO cells significantly restored the normal ciliary length and the normal localization of IFT88, mainly at the ciliary base (Fig 5, compare B with A; also see Fig 5G and 5H), although CCRK-EGFP itself displayed weak signals in the cytoplasm (see below). By contrast, CCRK(K33R)-EGFP did not significantly rescue the abnormal ciliary length or abnormal IFT88 enrichment at the ciliary tips (Fig 5C; also see Fig 5G and 5H), indicating that the kinase activity of CCRK is essential for its function. When the kinase domain construct CCRK(1–289) (Fig 1) was expressed in *CCRK*-KO cells, neither cilia elongation nor ciliary tip accumulation of IFT88 was rescued (Fig 5D; also see Fig 5G and 5H). Furthermore, and more importantly, the CCRK construct with only a 16-amino acid truncation from the C-terminus, CCRK(1–330), lacked the ability to rescue the abnormal phenotypes of *CCRK*-KO cells (Fig 5E; also see Fig 5G and 5H). These observations, together with the interaction data shown in Fig 1E and 1F, demonstrate that in addition to the kinase activity of CCRK, its interaction with BROMI is crucial for the function of CCRK in the regulation of ciliary protein trafficking.

As described above, CCRK-EGFP did not display distinct localization within cilia (Fig 5B), even though expression of CCRK-EGFP in CCRK-KO cells were confirmed by immunoblot analysis (Fig 5I); the results were consistent with a previous study on HA-tagged CCRK expressed in NIH3T3 cells [45]. However, after trying various fixation/permeabilization conditions, we were able to detect CCRK-EGFP signals at the base of cilia in a substantial population of cells by confocal microscopy (Fig 5J), although we were still unable to detect any signal within cilia.

In view of the above data showing that the kinase activity of CCRK is crucial for its function, we then analyzed whether the exogenous expression of ICK or its mutant at the TDY motif could recue the abnormal phenotypes of *CCRK*-KO cells, as ICK is phosphorylated by CCRK at the TDY motif [41,42]; however, our previous study showed that, when exogenously expressed in *ICK*-KO cells, ICK(WT), but not ICK(T157E) or ICK(T157A), which are a phosphomimetic mutant and a phosphorylation-defective mutant, respectively [42,56], rescued cilia elongation and IFT88 accumulation in the ciliary tips [30]. When mChe-fused ICK(WT), ICK(T157E), and ICK(T157A) were expressed in *CCRK*-KO cells, these proteins were found to be predominantly located at the ciliary tips (Fig 6A–6C), indicating that CCRK does not contribute to the anterograde trafficking of ICK to the ciliary tips. These observations are compatible with our previous study showing that ICK is transported to the tip via binding to the IFT-B complex [30]. However, neither ICK(WT), ICK(T157A), or ICK(T157E) substantially rescued cilia elongation or IFT88 tip accumulation in *CCRK*-KO cells (Fig 6A–6C; also see Fig 6I). When expressed in control RPE1 cells, ICK(T157E) and ICK(T157A) were substantially enriched at the ciliary tips, compared with ICK(WT) (Fig 6E–6G; also see Fig 6J); the ciliary tip enrichment of the ICK mutants was similar to that observed when these mutants were expressed in *ICK*-KO cells [30]. These observations suggest two possibilities. One is that although ICK undergoes phosphorylation by CCRK, it does not function downstream of CCRK, as ICK is mainly found at the tips of cilia, whereas CCRK is found at the ciliary base and does not demonstrate distinct localization within cilia (Fig 5B and 5J). It is therefore possible that at least some of CCRK functions can be mediated by its substrates other than ICK. The other possibility is that the phosphorylation-dephosphorylation cycle, rather than phosphorylation alone catalyzed by CCRK, is crucial for ICK function. We favor the latter possibility, in view of the facts that ICK(WT), but neither ICK(T157A) nor ICK(T157E) was able to rescue

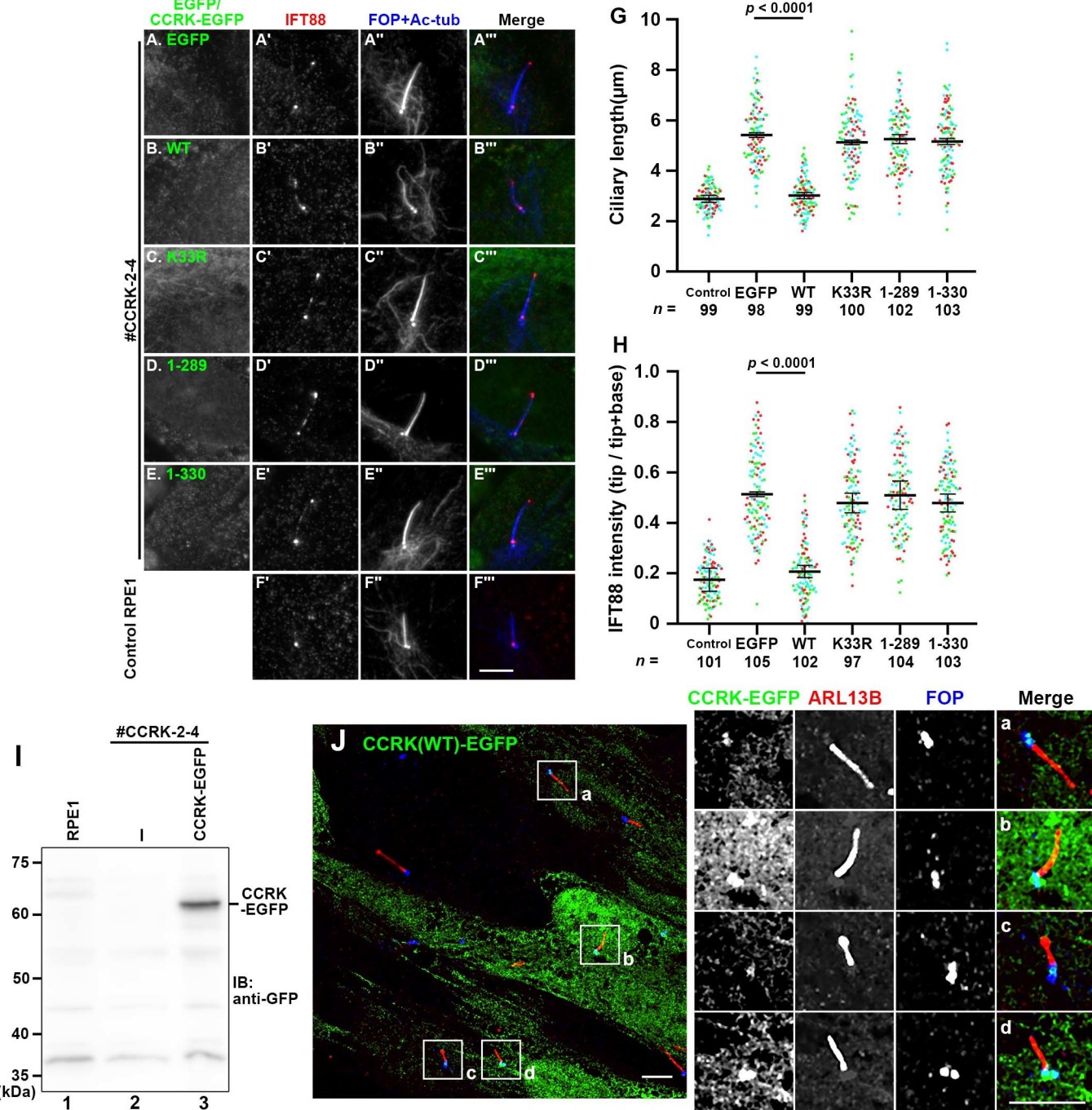

**Fig 5. Kinase activity of CCRK and its interaction with BROMI are required for its functions.** (A–E) *CCRK*-KO cells (#CCRK-2-4) stably expressing EGFP (A), or EGFP-fused CCRK(WT) (B), CCRK(K33R) (C), CCRK(1–289) (D), or CCRK(1–330) (E), or control RPE1 cells (F) were immunostained for EGFP (A–E), IFT88 (A′–F′), and FOP + Ac-tubulin (A″–F″). (G) Ciliary lengths of cells shown in (A″–F″) were measured and expressed as scatter plots. (H) The relative staining intensity of IFT88 at the ciliary tip and base in cells shown in (A′–F′) were estimated, and the intensity ratios of tip/(tip + base) were expressed as scatter plots. Different colored dots represent three independent experiments, horizontal lines indicate the means, and error bars are the SDs. Statistical significances were calculated using one-way ANOVA followed by the Dunnett's multiple comparison test. (I) Lysates prepared from control RPE1 cells (lane 1), *CCRK*-KO cells (#CCRK-2-4) (lane 2), or *CCRK*-KO cells stably expressing CCRK(WT)-EGFP (lane 3) were processed for immunoblotting analysis using anti-GFP antibody. (J) Confocal microscopy image of *CCRK*-KO cells (#CCRK-2-4) stably expressing CCRK(WT)-EGFP. The cells were fixed, permeabilized, incubated with anti-ARL13B and anti-FOP antibodies, and observed using a confocal microscope as described in Materials and methods. The boxed regions (a–d) were 2.5-fold magnified and shown at the right. Scale bars, 5 μm.

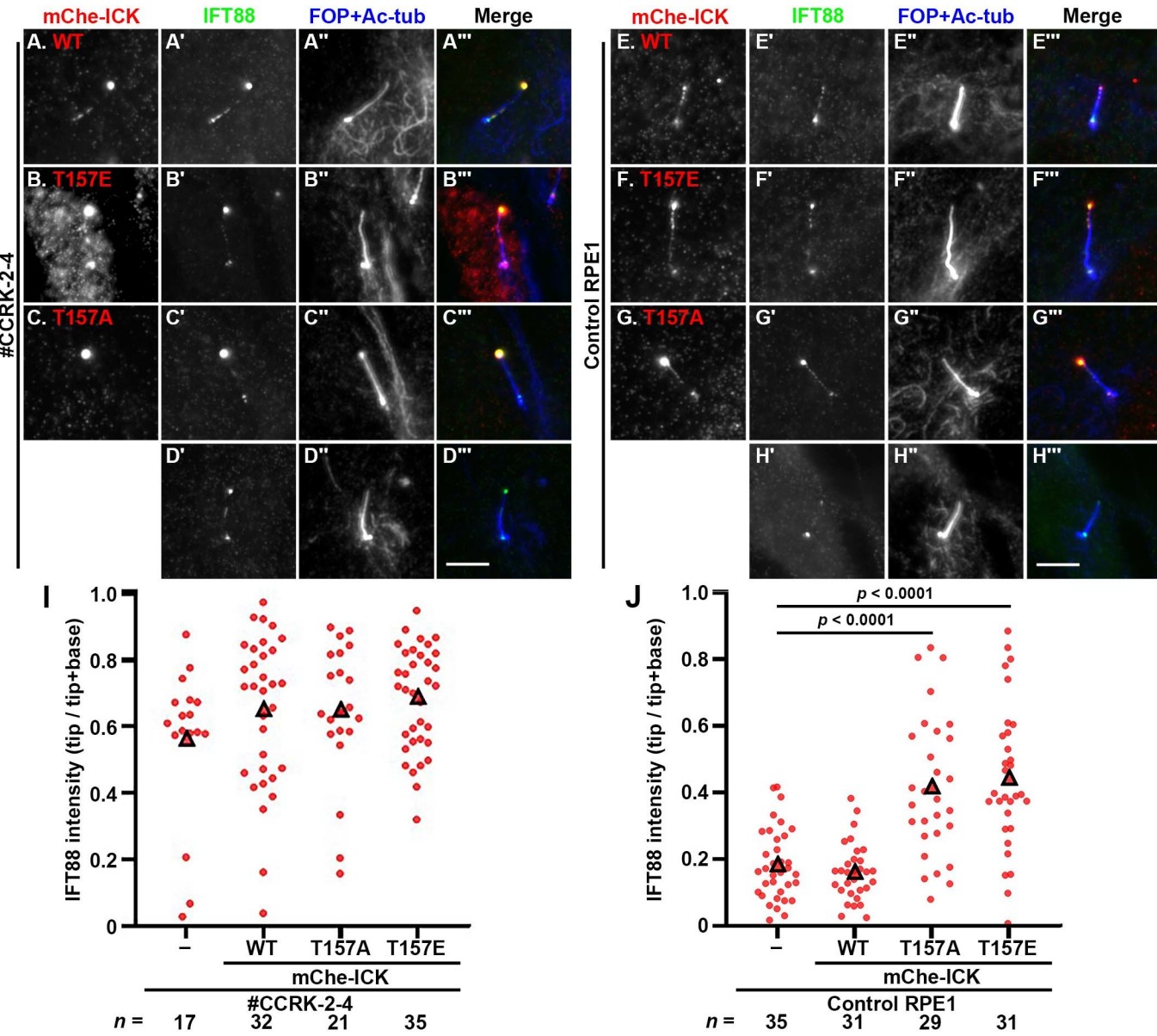

**Fig 6. ICK(WT) and its phosphorylation site mutants were unable to rescue the abnormal phenotypes of *CCRK*-KO cells.** *CCRK*-KO cells (#CCRK-2-4) (A–D) or control RPE1 cells (E–H) expressing mChe-fused ICK(WT) (A, E), ICK(T157E) (B, F), or ICK(T157A) (C, G) were immunostained for IFT88 (A′–H′) and FOP + Ac-tubulin (A″–H″). The original *CCRK*-KO cells and RPE1 cells were also stained for IFT88 and FOP + Ac-tubulin. Scale bars, 5 μm. (I, J) Relative staining intensities of IFT88 at the ciliary tips and base in the cells shown in (A)–(D) and (E)–(H), respectively, were analyzed as described in the legend for Fig 2N.

the abnormal ciliary phenotypes of *ICK*-KO cells, as reported previously [30], and that both ICK(T157A) and ICK(T157E) led to ciliary defects when expressed in control RPE1 cells (Fig 6F, 6G and 6J).

## Discussion

In a previous study, we showed that ICK is transported to the ciliary tips via its binding to the IFT machinery, where it then regulates the turnaround process at the tips by phosphorylating a substrate [30]. Although CCRK phosphorylates ICK at its TDY motif [41–43] and interacts

with BROMI [45], and mutations in both the homologs of CCRK and ICK are known to cause cilia elongation in a variety of organisms [31,37,43,44,48–56], the role of CCRK in ciliary protein trafficking has not been analyzed in detail nor compared with that of ICK in the same cell background. We here showed that the phenotypes of *CCRK*-KO RPE1 cells resemble those of *ICK*-KO RPE1 cells, although the abnormal phenotypes appear to be less robust in *CCRK*-KO cells than in *ICK*-KO cells. In *CCRK*-KO and *ICK*-KO cells, variations in ciliary length as well as average ciliary lengths are greater than those of control cells. Furthermore, not only the IFT-A and IFT-B complexes, but also the kinesin-II and dynein-2 motors are significantly enriched at the ciliary tips, and the bulged tip structures with the accumulated proteins are eliminated as ECVs (Figs 2 and 3). In addition, the exit of GPR161 and SMO from cilia was suppressed in *CCRK*-KO and *ICK*-KO cells, and rescue experiments of *CCRK*-KO cells with various CCRK constructs demonstrated that the kinase activity of CCRK and its interaction with BROMI are crucial for its function.

Previous studies using *Ccrk*-KO cells derived from IMCD3 mouse inner medullary collecting duct cells [44] and using mutant strains of CCRK homologs in *Chlamydomonas* [80] and in *C. elegans* [54] demonstrated that these mutant cilia/flagella show only small changes in IFT rates and frequencies, although cilia of *Ccrk*-KO IMCD3 cells demonstrated distal tip swelling and IFT88 accumulation at the tips [44]. However, we here showed the significant accumulation of components of the IFT machinery, including the kinesin-II and dynein-2 motor proteins, at the ciliary tips, and elimination of the accumulated proteins via ECVs in both *CCRK*-KO and *ICK*-KO cells, although the abnormal accumulation appeared to be more robust in *ICK*-KO cells than in *CCRK*-KO cells (Figs 2 and 3). These observations indicate that the slight slowdown somewhere in the IFT turnaround process, including disassembly and reassembly of the IFT machinery, motor switching, loading of retrograde cargos, and initiation of retrograde trafficking, in the absence of CCRK or ICK therefore might be responsible for the enrichment of IFT components at the ciliary tips. In this context, it is interesting to note previous studies demonstrating the fate of heterotrimeric kinesin-II after motor switching at the ciliary tips. In *C. elegans* sensory cilia and mouse olfactory cilia, kinesin-II-positive particles were reported to move in both the anterograde and retrograde directions [81–83], indicating that heterotrimeric kinesin-II undergoes retrograde IFT from the tip to the base of cilia after motor switching. On the other hand, a previous study on *Chlamydomonas* flagella suggested that kinesin-II is dissociated from the IFT machinery at the tip and simply diffuses back to the base of cilia [23]. Our results showing the significant enrichment of KAP3-EGFP at the ciliary tips in both *CCRK*-KO and *ICK*-KO cells indicates that at least some proportion of heterotrimeric kinesin-II undergoes retrograde trafficking after motor switching at the tips if the switching occurs normally.

A previous study in *C. elegans* showed that a CCRK ortholog as well as an ICK ortholog was found at the distal segments of sensory cilia [54], indicating that the CCRK ortholog can activate the ICK ortholog in *C. elegans* cilia. However, as CCRK was not found within cilia although it was detectable at the ciliary base in mammalian cells (Fig 5B and 5J) [45], there still remains the fundamental question as to how CCRK regulates the turnaround process. For the following reasons, it is likely that CCRK regulates ciliary protein trafficking via activating ICK or promoting its trafficking to the ciliary tips: CCRK phosphorylates ICK at its TDY motif [41,42]; phosphorylation of an ICK ortholog is not detected in a *Chlamydomonas* mutant strain of a CCRK ortholog [43]; and the abnormal phenotypes of *CCRK*-KO and *ICK*-KO cells closely resemble each other (Figs 2–4). However, a role of CCRK in promoting ICK trafficking is unlikely, because ICK can be transported to the ciliary tips of *CCRK*-KO cells as in control cells (Fig 6). Then, if the phosphorylation of ICK by CCRK is important in regulating ciliary protein trafficking, where does the phosphorylation occur? In view of the fact that ICK is

found at the ciliary base as well as being concentrated at the ciliary tips (see Fig 6E) [30] and CCRK is detectable at the ciliary base (Fig 5J), the most straightforward explanation is that ICK is phosphorylated by CCRK at the base, then is trafficked to the tips via binding to the IFT machinery. In line with this notion, previous superresolution imaging studies suggested that there is a potential waiting place for IFT particles at the ciliary base [84–86]. Otherwise, in view of the relatively small size of CCRK, in conjunction with the fact that the *C. elegans* CCRK ortholog is found within cilia [54], it can freely permeate the molecular sieve of the ciliary gate [87,88] and phosphorylate ICK within cilia. By elegant experiments using *Chlamydomonas* flagella, Nievergelt *et al.* has recently reported as a preprint that the conversion from anterograde to retrograde IFT trains does not require the stationary tip machinery and can occur in the middle of flagella [89], supporting the above possibility that ICK activation does not necessarily occur at the ciliary tip.

The data presented in this study (Fig 6) and those presented in our previous study [30] using the ICK(T157E) and ICK(T157A) mutants suggest that the phosphorylation-dephosphorylation cycle, rather than the phosphorylation of ICK, is important for its function in regulating the turnaround process at the ciliary tips, although a previous study using mouse NIH3T3 cells suggested that CCRK controls ciliary length simply by phosphorylating ICK [56]. In this context, it is interesting to note that previous proteomic analyses of *Chlamydomonas* flagella and mammalian cilia identified some phosphoprotein phosphatases (PPs) [90–92]. In addition, a previous study indicated that PP5 can interact with ICK and dephosphorylate ICK at T157 [41], although PP5 was not identified in the above cilia proteomic analyses [91,92]. Therefore, a future issue to be clarified is the mechanism of regulation of not only the phosphorylation of ICK by CCRK, but also its dephosphorylation.

Although the phenotypes of *CCRK*-KO and *ICK*-KO cells resemble each other (Figs 2–4), the facts that *Ccrk* mutant mouse embryos demonstrate earlier lethality than *Ick* mutant ones and that these mutant embryos show distinct morphological and patterning phenotypes [32,37,44] suggest that at least some of CCRK functions can be mediated by its substrates other than ICK. Therefore, identification of CCRK substrates other than ICK is also an important issue to be addressed in the future.

How BROMI is involved in the regulation of the CCRK function is also an important future issue. As shown in Fig 1, BROMI interacts with the C-terminal noncatalytic region of CCRK. A previous study suggested that BROMI stabilizes CCRK as the CCRK level was reduced in *Bromi* mutant cells in comparison to wild-type cells [45]. The same study also showed that *Bromi* mutant neuroepithelial cilia had abnormal morphology [45]. We here showed that a CCRK construct [CCRK(1–330)] defective in BROMI binding as well as its kinase-dead mutant was not able to rescue the abnormal phenotype of *CCRK*-KO cells (Fig 5). Therefore, it is also possible that BROMI contributes to activation and/or substrate specificity of CCRK/CDK20, like the relationships between CDKs and cyclins.

## Supporting information

**S1 Fig. Genomic PCR and sequence analyses of the *CCRK*-KO cell lines.** (A, B) Genomic DNAs were extracted from the *CCRK*-KO cell lines #CCRK-1-1 (A) and #CCRK-2-4 (B), which were established using a donor knockin vector containing distinct target sequences, and subjected to PCR analysis using the indicated primer sets (see S2 Table) to detect alleles with a small indel or no insertion (a, a'), or with forward (b, b') or reverse (c, c') integration of the donor knockin vector. M, molecular weight markers (PSU1 + ladder). (C, D) Alignments of allele sequences of the #CCRK-1-1 (C) and #CCRK2-4 (D) cell lines determined by sequencing of the PCR products shown in (A) and (B). Positions of the target sequence and protospacer

adjacent motif (PAM) sequence, and insertion sites and directions of the donor knockin vector are indicated.
(JPG)

**S2 Fig.**
(PPTX)

**S1 Table. Plasmids used in this study.**
(PDF)

**S2 Table. Antibodies used in this study.**
(PDF)

**S3 Table. Oligo DNAs used in this study.**
(PDF)

**S1 Video. ECV formation in a *CCRK*-KO cell (from Fig 2P).** *CCRK*-KO cells (#CCRK-2-4) stably expressing ARL13B(ΔGD)-EGFP were processed for live cell imaging, as described in Materials and methods.
(MOV)

**S2 Video. ECV formation in a *CCRK*-KO cell (from Fig 2Q).** *CCRK*-KO cells (#CCRK-2-4) stably expressing ARL13B(ΔGD)-EGFP were processed for live cell imaging as described in Materials and methods.
(MOV)

## Acknowledgments

We thank Peter McPherson for providing the plasmids for lentiviral production, and Takahisa Furukawa for providing the ICK plasmid, and Helena Akiko Popiel for critical reading of the manuscript.

## Author Contributions

**Conceptualization:** Yohei Katoh, Kazuhisa Nakayama.

**Formal analysis:** Tatsuro Noguchi, Kentaro Nakamura, Yuuki Satoda.

**Funding acquisition:** Yohei Katoh, Kazuhisa Nakayama.

**Investigation:** Tatsuro Noguchi, Kentaro Nakamura, Yuuki Satoda.

**Project administration:** Yohei Katoh, Kazuhisa Nakayama.

**Supervision:** Yohei Katoh, Kazuhisa Nakayama.

**Validation:** Tatsuro Noguchi, Kentaro Nakamura, Kazuhisa Nakayama.

**Writing – original draft:** Kazuhisa Nakayama.

**Writing – review & editing:** Tatsuro Noguchi, Kentaro Nakamura, Yuuki Satoda, Yohei Katoh, Kazuhisa Nakayama.

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
