## [Decision Letter · Decision Letter 0]

13 Jul 2021

PONE-D-21-18831

CCRK/CDK20 regulates ciliary retrograde protein trafficking via interacting with BROMI/TBC1D32

PLOS ONE

Dear Dr. Nakayama,

Thank you for submitting your manuscript to PLOS ONE. After careful consideration by 3 reviewers, we feel that the manuscript needs some improvements before it can be accepted. Therefore, we invite you to submit a revised version of the manuscript that addresses the points raised during the review process.

We look forward to receiving your revised manuscript.

Kind regards,

Oliver E Blacque

Academic Editor

PLOS ONE

Journal Requirements:

Reviewers' comments:

Reviewer's Responses to Questions

**Comments to the Author**

1. Is the manuscript technically sound, and do the data support the conclusions?

Reviewer #1: Yes

Reviewer #2: Yes

Reviewer #3: Partly

2. Has the statistical analysis been performed appropriately and rigorously? 

Reviewer #1: Yes

Reviewer #2: Yes

Reviewer #3: Yes

3. Have the authors made all data underlying the findings in their manuscript fully available?

Reviewer #1: Yes

Reviewer #2: Yes

Reviewer #3: Yes

4. Is the manuscript presented in an intelligible fashion and written in standard English?

Reviewer #1: Yes

Reviewer #2: Yes

Reviewer #3: Yes

5. Review Comments to the Author

Reviewer #1: In this manuscript, Noguchi et al report that CCRK/CDK20 kinase regulates ciliary retrograde protein trafficking by interacting with BROMI. Using in vitro binding assays, mutant phenotype analysis, and cell imaging, the authors show that CCRK binds to BROMI via its C-terminal tail and this binding ability is essential for its ciliary function. The authors also report that Ccrk mutant phenotypes resemble those in Ick mutants and propose that CCRK functions by activating ICK. Although mutant phenotypes of Ccrk and Ick are confirmatory, functional characterization of the BROMI-binding C-terminus is informative and interesting which expands the current knowledge about how CCRK regulates ciliogenesis.

In general, the performed experiments are explicitly described and the conclusions are mostly well supported. Before considering for its publication, I have several minor questions.

1. The authors claim that CCRK regulates ciliary retrograde protein trafficking. However, as mentioned by the authors in the manuscript and abstract, the accumulation of ciliary proteins inside Ccrk mutant cilia could be a result of either impaired protein transport or IFT switching. Previous studies (Broekhuis et al. 2014; Yi et al. 2018) show that retrograde speed is not affected in Ick or Ccrk mutants. It is likely that CCRK/ICK play more important roles in IFT reorganization, motor switching, or cargo loading, other than regulating transport process (trafficking). In the manuscript, the authors didn't provide evidence to distinguish these two possibilities. I suggest the authors tuning down their claim, for example, by using phrases such as "protein recycling" or "ciliary exit".

2. The authors didn't mention the rationale of generating two truncation forms of BROMI in Figure 1B (column 5 and 6) and didn't draw a conclusion on that. I suggest the authors providing a list in Figure 1B to indicate which fragment of BROMI is required for its binding to CCRK.

3. Expression of CCRK-EGFP rescued Ccrk mutant phenotypes, but the expressed protein is not visible. Have the authors tried immunostaining to check CCRK localization? Previous in vitro and in vivo studies all indicate that CCRK may function by phosphorylating and activating ICK/MAK. However, ICK/MAK localizes in cilia, which is different to CCRK/LF2p found in Chlamydomonas (Tam et al. 2007) and mammalian cells (Ko et al. 2010) (in the cytoplasm). In C. elegans sensory cilia, CCRK and ICK homologs are both enriched in cilia (Yi et al. 2018). It would be valuable to clarify the mechanism of CCRK/ICK/MAK pathway in different systems if the authors could further check the localization pattern of CCRK.

4. Expression of any of the three forms of ICK (WT, T157A, T157E) didn't rescue CCRK phenotypes. Is it possible that MAK but not ICK is involved in the tested cells?

5. Page 7 paragraph 1 "truncation of the first 16-amino acids...", I guess the authors mean the last 16-amino acids?

6. Is the tail domain that binds to BROMI conserved or not?

Reviewer #2: CCRK has emerged in the last years to regulate cilia length controlling axonemal microtubule stability in C. elegans. Moreover, it has been shown that CCRK mice mutant cells present defective regulation of ciliary length, morphology and intraflagellar transport. In this study, the authors expand the study of CCRK function in RPE1 cells in parallel with the study of ICK kinase, a phosphorylation target of CCRK also involved in controlling cilia length and IFT trafficking. The authors show that CCRK-KO phenotype resemble that of the ICK-KO cells, with accumulation of IFT particles at the bulged ciliary tips which were eliminated as extracellular vesicles. They also take advantage of the visible immunoprecipitation (VIP) assay, previously developed by Nakayama and colleagues, to characterise the CCRK interaction with BROMI. They found that the last CCRK 16 amino acids are essential in BROMI binding and this CCRK defective mutant is not able to rescue CCRK-KO cells phenotype, suggesting that the interaction with BROMI is essential for CCRK function in the cilia. These finding are of interest to the molecular and cell biology communities. The experiments were performed at high standards and the manuscript is well organised and written clearly.

Points for improvement:

1.The CCRK KO cell lines were established using CRISPOR. Disruption of CCRK alleles was confirmed by genomic sequencing. It would be informative to check protein expression by immunoblot as well if possible.

2.When comparing phenotypes in CCRK and ICK KO cells of Kap3 accumulation at the tip with those in C. elegans the authors should be careful to mention that in C. elegans the trafficking system is different and two kinesin motors exist.

3.In the result section the authors state that accumulation of Kap3 and DYNC2LI1 suggest that motor switching did not occur at the distal tips in the absence of CCRK or ICK. It is unlikely that motor switching is totally impaired because mutations in motors proteins cause much severe phenotypes with loss of cilia extension.

4. It is known that ICK is phosphorylated by CCRK and both KO cell lines show similar phenotypes, but a direct link has not been established in this paper. Neither ICK wild type or the phosphomimetic and phosphorylation-defective mutants can rescue the CCRK-KO phenotype. The authors suggest that the phosphorylation/dephosphorylation cycle it is important rather than phosphorylation or dephosphorylation of ICK alone, which is plausible. I wonder if they might speculate about of the identity of the phosphatase involved in the dephosphorylation of ICK in the discussion.

5.The authors show that ICK is localised mainly at the cilia tip while overexpressed CCRK GFP tagged construct was not observed localising in cilia of CCRK KO cells. Why is the localisation in Crtl wild type cells not shown? Could the authors check the localisation of the endogenous proteins?

6.Authors show that the last 16 amino acids of CCRK are essential for interaction with BROMI and CCRK defective in BROMI binding could not rescue ciliary abnormal phenotypes in CCRK KO cells. What is the function of BROMI and when does this interaction occur? It has been shown previously that neural progenitor BROMI cilia are abnormally shaped, but it would be interesting to see if the phenotype of BROMI defective cells resemble that of CCRK-KO cells.

Reviewer #3: The work extends our understanding of CCRK function in the control of ciliogenesis although many questions remain. Below are some recommendations for improving the manuscript:

1.Minor: I would recommend revising the first part of the abstract for flow so that the reader can follow more easily. For example, “On the other hand” in the second sentence only confuses things and should be omitted.

2.Minor: The authors should discuss the how the effects of CCRK disruption on ciliary length is manifested in different contexts: i.e., partial knockdown in mouse NIH3T3 fibroblasts results in ciliary elongation (Yang et al EMBO reports 2013), complete loss of CCRK in mouse embryonic fibroblasts results in a broader range of ciliary lengths (such that the average length is not statistically different from that of wt cells—Snouffer et al., PLoS Genet., 2017), and complete loss in human RPE1 cells results in statistically-significant ciliary elongation (the current study).

3.Minor: The authors make the claim, “Our results showed that CCRK-KO cells accumulated IFT proteins at their bulged ciliary tips, which were eliminated as ECVs”. While the first part of this statement is supported, the authors do not actually show that accumulated IFT proteins are removed from the cilium in ECV’s as they only monitored Arl13b-GFP fluorescence moving away in ECVs (but not IFT88 fluorescence) in the movies. Moreover, the authors should make clear whether ECV shedding occurs in wild-type cells and if so, is the rate of shedding and size of ECVs similar or different from Ccrk mutant cells.

4.Minor: The authors offer two explanations for their findings on the ciliary tip accumulation of IFT88, KAP3, and DYNC2LI1 in the statement: “One is that motor switching did not occur at the distal tips in the absence of CCRK or ICK, and the other is that, after motor switching, both the kinesin and the dynein motors were accumulated at the tips owing to impaired retrograde trafficking of the IFT machinery in the absence of CCRK or ICK”. To push this a bit further, it would have been good to see an actual measurement of IFT in Ccrk mutant cells (given that the authors have already shown they can do live imaging of Ccrk mutant cilia). Perhaps it should be noted that when Snouffer et al (2017) imaged IFT in Ccrk mutant IMCD3 cells, they observed relatively minor effects of the mutation on IFT88-YFP anterograde and retrograde transport (both rate and frequency), although such cells showed the same IFT88 accumulation at ciliary tip phenotype as observed in the current study.

5.Major: The findings in Figure 6 (ICK variants in Ccrk mutant and control cells) should be quantitated and analyzed statistically (as performed for other data in the manuscript). That said, the findings are a bit surprising and should be discussed in light of other data. Previous work showed that overexpression of WT, but not TDY mutant ICK, shortens/obliterates cilia in NIH3T3 cells, suggesting that phosphorylation of ICK by CCRK of T157 is important for its activity (Moon et al. PNAS, 2014). Moreover, Yang et al (EMBO Reports 2013) similarly showed that overexpression of WT ICK shortens cilia in control NIH3T3 cells, but there is a minimal effect in cells when Ccrk was knocked down by siRNAs in such cells. In addition, overexpression of a T157A mutant in control or Ccrk KD cells did not result in significant ciliary shortening, whereas overexpression of a phosphomimetic form (T154E) shortened cilia in both control and Ccrk knockdown cells. Finally, the authors of the current manuscript previously showed that a WT but not T157A ICK form could rescue the Ift88 tip accumulation phenotype seen in Ick mutant RPE1 cells (Nakamura et al., JCB 2020). Collectively, these data strongly argue against the possibility (which not favored by the authors) that “although ICK undergoes phosphorylation by CCRK, it does not function downstream of CCRK, as ICK is mainly found at the tips of cilia, whereas CCRK does not demonstrate distinct ciliary localization (Fig. 5B)”. That said, the mere fact that Ccrk mutant mouse embryos show earlier lethality as well as a distinct morphological and patterning phenotype when compared to Ick mutant embryos (Snouffer et al PLoS Genet., 2017, Chaya et al. EMBO J, 2014; Moon et al. PNAS, 2014), suggests that at least some of CCRK’s functions are independent of ICK, likely mediated by other CCRK substrates. For example, ciliary shortening functions of ICK may depend on CCRK activity (e.g., Yang et al 2013), whereas the data in Figure 6 suggests that the control of IFT protein accumulation at ciliary tip by CCRK is mediated by different mediators. These data do not rule out the possibility that phosphorylation of ICK by CCRK must be dynamic (toggling) for proper ICK function, but they do point to an even stronger model, which was not suggested by the authors, that CCRK functions are mediated by both ICK-dependent and ICK-independent mechanisms. The manuscript would be strengthened considerably by discussing existing data that favor ICK acting downstream of CCRK, but also by discussing the possibility of ICK-dependent and independent roles for ICK.

6. PLOS authors have the option to publish the peer review history of their article (what does this mean?). If published, this will include your full peer review and any attached files.

Reviewer #1: No

Reviewer #2: No

Reviewer #3: No

---

## [Author Response · Author response to Decision Letter 0]

21 Aug 2021

Response to Reviewer #1

1. The authors claim that CCRK regulates ciliary retrograde protein trafficking. However, as mentioned by the authors in the manuscript and abstract, the accumulation of ciliary proteins inside Ccrk mutant cilia could be a result of either impaired protein transport or IFT switching. Previous studies (Broekhuis et al. 2014; Yi et al. 2018) show that retrograde speed is not affected in Ick or Ccrk mutants. It is likely that CCRK/ICK play more important roles in IFT reorganization, motor switching, or cargo loading, other than regulating transport process (trafficking). In the manuscript, the authors didn't provide evidence to distinguish these two possibilities. I suggest the authors tuning down their claim, for example, by using phrases such as "protein recycling" or "ciliary exit".

According to the comment of the reviewer, we have toned down the claim by mentioning that various events (disassembly and reassembly of the IFT machinery, motor switching, loading of retrograde cargos, and initiation of retrograde trafficking) in the turnaround process could be affected in the absence of CCRK or ICK in DISCUSSION (page 11, 2nd paragraph). 

2. The authors didn't mention the rationale of generating two truncation forms of BROMI in Figure 1B (column 5 and 6) and didn't draw a conclusion on that. I suggest the authors providing a list in Figure 1B to indicate which fragment of BROMI is required for its binding to CCRK.

We think that the reviewer probably misspelled Figure 1D (columns 5 and 6) as Figure 1B (columns 5 and 6). We have rewritten the text describing Figs 1D and 1E (Figs. 1G and 1H in the revised MS) (page 7, 3rd paragraph), and added new Fig. 1D (the predicted 3D structure of BROMI in the AlphaFold Protein Structure Database) to supplement this description.

3. Expression of CCRK-EGFP rescued Ccrk mutant phenotypes, but the expressed protein is not visible. Have the authors tried immunostaining to check CCRK localization? Previous in vitro and in vivo studies all indicate that CCRK may function by phosphorylating and activating ICK/MAK. However, ICK/MAK localizes in cilia, which is different to CCRK/LF2p found in Chlamydomonas (Tam et al. 2007) and mammalian cells (Ko et al. 2010) (in the cytoplasm). In C. elegans sensory cilia, CCRK and ICK homologs are both enriched in cilia (Yi et al. 2018). It would be valuable to clarify the mechanism of CCRK/ICK/MAK pathway in different systems if the authors could further check the localization pattern of CCRK.

According to the valuable comment of the reviewer, we have tried a variety of fixation/permeabilization/staining conditions and found, by confocal microscopy, that CCRK-EGFP is often localized to the ciliary base in the background of the cytoplasmic staining. We have therefore added the new data as Fig. 5J and consequently a new paragraph in RESULTS (page 10, 2nd paragraph), and discussed the localization of CCRK in relation to its function in DISCUSSION (the paragraph spanning pages 11 and 12).

4. Expression of any of the three forms of ICK (WT, T157A, T157E) didn't rescue CCRK phenotypes. Is it possible that MAK but not ICK is involved in the tested cells?

Although MAK is expressed in restricted tissues and was shown to be expressed in retinal photoreceptor cells but not in retinal pigment epithelial (RPE) cells (see Ref. 31), we have discussed the possibility that the abnormal phenotype of CCRK-KO RPE1 cells may be attributable to the lack of phosphorylation of CCRK substrates other than ICK, including MAK in the revised MS (page 10, the last two sentences of the “RESULTS” section). 

5. Page 7 paragraph 1 "truncation of the first 16-amino acids...", I guess the authors mean the last 16-amino acids?

"the first 16-amino acids" has been changed to "the last 16-amino acids" in the revised MS (page 7, 2nd paragraph).

6. Is the tail domain that binds to BROMI conserved or not?

New Fig. 1B has been added to compare the sequences of the vertebrate CCRK tail region, and new Fig. 1C showing the predicted 3D structure of CCRK in the AlphaFold Protein Structure Database has been added. In accordance, a new sentence has been added (page 7, the last sentence in the 2nd paragraph).

Response to Reviewer #2

1.The CCRK KO cell lines were established using CRISPOR. Disruption of CCRK alleles was confirmed by genomic sequencing. It would be informative to check protein expression by immunoblot as well if possible.

We have performed immunoblotting analysis using commercially available anti-CCRK antibody [Abcam, #EPR7338(2)] on the same blot as that shown in new Fig. 5I, but have not detected a specific band on neither the control RPE1 lane nor the lane of CCRK-KO cells expressing CCRK-EGFP.

2.When comparing phenotypes in CCRK and ICK KO cells of Kap3 accumulation at the tip with those in C. elegans the authors should be careful to mention that in C. elegans the trafficking system is different and two kinesin motors exist.

We have mentioned that two types of kinesin-2 are involved in intraciliary trafficking in C. elegans (page 9, 1st paragraph).

3.In the result section the authors state that accumulation of Kap3 and DYNC2LI1 suggest that motor switching did not occur at the distal tips in the absence of CCRK or ICK. It is unlikely that motor switching is totally impaired because mutations in motors proteins cause much severe phenotypes with loss of cilia extension.

We have changed the sentence pointed by the reviewer from “motor switching did not occur at the distal tips in the absence of CCRK or ICK” to “motor switching at the distal tips was compromised, at least partially, in the absence of CCRK or ICK”. Furthermore, according to the comment 1 of Reviewer 1, we have discussed the possibility that various events that occur in the turnaround process may be affected in the absence of CCRK or ICK, and we only mentioned motor switching as one of such possible events (see response to the comment 1 of Reviewer 1).

4. It is known that ICK is phosphorylated by CCRK and both KO cell lines show similar phenotypes, but a direct link has not been established in this paper. Neither ICK wild type or the phosphomimetic and phosphorylation-defective mutants can rescue the CCRK-KO phenotype. The authors suggest that the phosphorylation/dephosphorylation cycle it is important rather than phosphorylation or dephosphorylation of ICK alone, which is plausible. I wonder if they might speculate about of the identity of the phosphatase involved in the dephosphorylation of ICK in the discussion.

Although we had already discussed phosphoprotein phosphatases that may function within cilia in the unrevised MS, we have taken the discussion further in the revised MS (page 11, 2nd paragraph, the last three sentences).

5.The authors show that ICK is localised mainly at the cilia tip while overexpressed CCRK GFP tagged construct was not observed localising in cilia of CCRK KO cells. Why is the localisation in Crtl wild type cells not shown? Could the authors check the localisation of the endogenous proteins?

We have performed immunofluorescence analysis using commercially available anti-CCRK antibody (Abcam) but have not detected specific signals. Instead, as described in the response to Comment 3 of Reviewer #1, we have found that CCRK-EGFP is often localized to the ciliary base in the background of the cytoplasmic staining by confocal microscopy (new Fig. 5J).

6.Authors show that the last 16 amino acids of CCRK are essential for interaction with BROMI and CCRK defective in BROMI binding could not rescue ciliary abnormal phenotypes in CCRK KO cells. What is the function of BROMI and when does this interaction occur? It has been shown previously that neural progenitor BROMI cilia are abnormally shaped, but it would be interesting to see if the phenotype of BROMI defective cells resemble that of CCRK-KO cells.

We do not exactly know how BROMI is involved in the regulation of the CCRK function. Nevertheless, we have enhanced our discussion on this point, with reference to the relationships between CDKs and cyclins, (page 12, the last paragraph in DISCUSSION). 

Response to Reviewer #3

1.Minor: I would recommend revising the first part of the abstract for flow so that the reader can follow more easily. For example, “On the other hand” in the second sentence only confuses things and should be omitted.

By omitting “On the other hand”, we have rewritten the first part of the Abstract.

2.Minor: The authors should discuss the how the effects of CCRK disruption on ciliary length is manifested in different contexts: i.e., partial knockdown in mouse NIH3T3 fibroblasts results in ciliary elongation (Yang et al EMBO reports 2013), complete loss of CCRK in mouse embryonic fibroblasts results in a broader range of ciliary lengths (such that the average length is not statistically different from that of wt cells—Snouffer et al., PLoS Genet., 2017), and complete loss in human RPE1 cells results in statistically-significant ciliary elongation (the current study).

Although we had already mentioned that the variations in ciliary length appeared to be greater in CCRK-KO cells than in control cells in the unrevised MS, we have elaborated on this point further by citing Snouffer et al. (2017) in the revised MS (page 8, 1st paragraph, the last two sentences).

3.Minor: The authors make the claim, “Our results showed that CCRK-KO cells accumulated IFT proteins at their bulged ciliary tips, which were eliminated as ECVs”. While the first part of this statement is supported, the authors do not actually show that accumulated IFT proteins are removed from the cilium in ECV’s as they only monitored Arl13b-GFP fluorescence moving away in ECVs (but not IFT88 fluorescence) in the movies. Moreover, the authors should make clear whether ECV shedding occurs in wild-type cells and if so, is the rate of shedding and size of ECVs similar or different from Ccrk mutant cells.

There is a technical difficulty in addressing this point made by the reviewer. Namely, attempts to exogenously express two different fluorescent fusion proteins (e.g. ARL13B-EGFP and mCherry-IFT88) in KO cells often result in cell death for unknown reasons. In our previous study (Ref. 30), exogenous expression of ARL13B-EGFP and mCherry-IFT88s in ICK-KO cells was successfully done with various attempts. However, we expect that it will take a lot of time to try to do the same in CCRK-KO cells. To circumvent this problem, we detected the presence of punctate structures, which are positive for both IFT88 and ARL13B but negative for the basal body marker FOP in CCRK-KO cells as well as in ICK-KO cells. As shownd in our previous study on ICK-KO cells (Ref. 30), these punctate structures represent extracellular vesicles released from ciliary tips. In the revised MS, we have added new Fig. 2, P–R, to show these punctate structures, and accordingly added new sentences to explain these figures (page 8, 3rd paragraph).

4.Minor: The authors offer two explanations for their findings on the ciliary tip accumulation of IFT88, KAP3, and DYNC2LI1 in the statement: “One is that motor switching did not occur at the distal tips in the absence of CCRK or ICK, and the other is that, after motor switching, both the kinesin and the dynein motors were accumulated at the tips owing to impaired retrograde trafficking of the IFT machinery in the absence of CCRK or ICK”. To push this a bit further, it would have been good to see an actual measurement of IFT in Ccrk mutant cells (given that the authors have already shown they can do live imaging of Ccrk mutant cilia). Perhaps it should be noted that when Snouffer et al (2017) imaged IFT in Ccrk mutant IMCD3 cells, they observed relatively minor effects of the mutation on IFT88-YFP anterograde and retrograde transport (both rate and frequency), although such cells showed the same IFT88 accumulation at ciliary tip phenotype as observed in the current study.

This comment points out almost the same as the comment 1 of Reviewer 1, but the way of suggestion is different between each other. According to the comment of Reviewer 1, we have toned down the description on the role of CCRK and ICK in the turnaround process in the revised MS (see response to the comment 1 of Reviewer 1). Even so, we have cited the paper of Snouffer et al. (2017) and referred to tip swelling and IFT88 accumulation at the tips of cilia of CCRK-KO IMCD3 cells (page 11, 2nd paragraph).

5.Major: The findings in Figure 6 (ICK variants in Ccrk mutant and control cells) should be quantitated and analyzed statistically (as performed for other data in the manuscript). That said, the findings are a bit surprising and should be discussed in light of other data. Previous work showed that overexpression of WT, but not TDY mutant ICK, shortens/obliterates cilia in NIH3T3 cells, suggesting that phosphorylation of ICK by CCRK of T157 is important for its activity (Moon et al. PNAS, 2014). Moreover, Yang et al (EMBO Reports 2013) similarly showed that overexpression of WT ICK shortens cilia in control NIH3T3 cells, but there is a minimal effect in cells when Ccrk was knocked down by siRNAs in such cells. In addition, overexpression of a T157A mutant in control or Ccrk KD cells did not result in significant ciliary shortening, whereas overexpression of a phosphomimetic form (T154E) shortened cilia in both control and Ccrk knockdown cells. Finally, the authors of the current manuscript previously showed that a WT but not T157A ICK form could rescue the Ift88 tip accumulation phenotype seen in Ick mutant RPE1 cells (Nakamura et al., JCB 2020). Collectively, these data strongly argue against the possibility (which not favored by the authors) that “although ICK undergoes phosphorylation by CCRK, it does not function downstream of CCRK, as ICK is mainly found at the tips of cilia, whereas CCRK does not demonstrate distinct ciliary localization (Fig. 5B)”. That said, the mere fact that Ccrk mutant mouse embryos show earlier lethality as well as a distinct morphological and patterning phenotype when compared to Ick mutant embryos (Snouffer et al PLoS Genet., 2017, Chaya et al. EMBO J, 2014; Moon et al. PNAS, 2014), suggests that at least some of CCRK’s functions are independent of ICK, likely mediated by other CCRK substrates. For example, ciliary shortening functions of ICK may depend on CCRK activity (e.g., Yang et al 2013), whereas the data in Figure 6 suggests that the control of IFT protein accumulation at ciliary tip by CCRK is mediated by different mediators. These data do not rule out the possibility that phosphorylation of ICK by CCRK must be dynamic (toggling) for proper ICK function, but they do point to an even stronger model, which was not suggested by the authors, that CCRK functions are mediated by both ICK-dependent and ICK-independent mechanisms. The manuscript would be strengthened considerably by discussing existing data that favor ICK acting downstream of CCRK, but also by discussing the possibility of ICK-dependent and independent roles for ICK.

We have added new Figs. 6I and 6J for quantification and statistical analysis of the data shown in Fig. 6A-D and E–H. In addition, according to the point by the reviewer, we have discussed the possibility that CCRK functions by not only ICK-dependent but also ICK-independent mechanism by adding a new paragraph (page 12, the penultimate paragraph in DISCUSSION).

---

## [Decision Letter · Decision Letter 1]

24 Sep 2021

PONE-D-21-18831R1CCRK/CDK20 regulates ciliary retrograde protein trafficking via interacting with BROMI/TBC1D32PLOS ONE

Dear Dr. Nakayama,

I apologize for the delay in deciding on your revised manuscript. The reviewers are satisfied that you have addressed their concerns apart from a couple of minor concerns from reviewer 2 that may require a few small text changes in the manuscript. Once you respond I can make the final accept decision without going back to the reviewers, and ensure that your paper enters post acceptance production quickly.

Regards

Oliver

Reviewers' comments:

Reviewer's Responses to Questions

**Comments to the Author**

1. If the authors have adequately addressed your comments raised in a previous round of review and you feel that this manuscript is now acceptable for publication, you may indicate that here to bypass the “Comments to the Author” section, enter your conflict of interest statement in the “Confidential to Editor” section, and submit your "Accept" recommendation.

Reviewer #1: All comments have been addressed

Reviewer #3: (No Response)

2. Is the manuscript technically sound, and do the data support the conclusions?

Reviewer #1: Yes

Reviewer #3: Yes

3. Has the statistical analysis been performed appropriately and rigorously? 

Reviewer #1: Yes

Reviewer #3: Yes

4. Have the authors made all data underlying the findings in their manuscript fully available?

Reviewer #1: Yes

Reviewer #3: Yes

5. Is the manuscript presented in an intelligible fashion and written in standard English?

Reviewer #1: Yes

Reviewer #3: Yes

6. Review Comments to the Author

Reviewer #1: The authors have addressed all my concerns. I believe the manuscript is substantially improved and meets the criteria for publication in PLos One.

Reviewer #3: The authors have significantly improved the manuscript and I only have a few comments:

In the abstract (“...the overaccumulation of IFT proteins at the bulged ciliary tips, which are eliminated as extracellular vesicles….”) and on Pg. 4 (“Our results showed that CCRK-KO cells accumulated IFT proteins at their bulged ciliary tips, which were eliminated as ECVs.”) the wording is still a bit strong since the conclusion is based on detection of Ift88+ punctate structures rather than live imaging of Ift88-GFP. Instead, I recommend using the wording "... appear to be eliminated as...." to soften the claim.

On Pg. 10 and 12, the authors write:

“The second possibility is that the abnormal phenotype of CCRK-KO cells may be attributable to the lack of phosphorylation of CCRK substrates other than ICK, including MAK, although MAK is expressed in limited tissues [31, 79].” and “The candidate substrates include MAK, a tissue-specific paralog of ICK [31], and MOK, a more divergent paralog that is also expressed in a tissue-specific manner [79], both of which have a T(D/E)Y sequence. However, in view of the fact that Mak null mice are viable [93], it is likely that ICK, which is ubiquitously expressed [79], generally contributes to the regulation of ciliary functions."

Since MAK does not appear to be expressed in RPE cells and Mak mouse mutants are viable with the only known phenotype being in photoreceptor cells, it is probably not a good idea to mention MAK as an alternative potential mediator of CCRK’s functions (so, I’d remove it). In addition, it is not clear whether MOK expressed in RPE cells. If not, I would avoid mentioning MOK as a candidate mediator as well. Please be aware that CCRK substrates/mediators don’t necessarily have to be members of the ICK/MAK/MOK family.

Considering this second possibility put forward by the authors, they should consider that different ciliary functions of CCRK could be mediated by more than one factor and that the roles of CCRK mediators may vary depending on cellular context. In human RPE cells, it could be argued that many or all of CCRK’s functions are mediated by (still unidentified) factors other than ICK, given that forced ICK-T157E expression does not rescue the IFT88 tip accumulation phenotype of Ccrk mutants. However, in mouse (NIH3T3) fibroblasts, it appears that CCRK controls ciliary length simply by phosphorylating/activating ICK and toggling of ICK between phospho states is not important. This is supported by the fact that forced expression of ICK-T157E, but not wild-type or T157A ICK, is able to rescue the ciliary length phenotype of Ccrk knockdown NIH3T3 cells (Yang et al, EMBO reports 2013).

7. PLOS authors have the option to publish the peer review history of their article (what does this mean?). If published, this will include your full peer review and any attached files.

Reviewer #1: No

Reviewer #3: No

---

## [Author Response · Author response to Decision Letter 1]

26 Sep 2021

Response to the comments of Reviewer #3: 

In the abstract (“...the overaccumulation of IFT proteins at the bulged ciliary tips, which are eliminated as extracellular vesicles….”) and on Pg. 4 (“Our results showed that CCRK-KO cells accumulated IFT proteins at their bulged ciliary tips, which were eliminated as ECVs.”) the wording is still a bit strong since the conclusion is based on detection of Ift88+ punctate structures rather than live imaging of Ift88-GFP. Instead, I recommend using the wording "... appear to be eliminated as...." to soften the claim.

According to this comment, we have changed the wording to “which appear to be eliminated as extracellular vesicles,” in Abstract and “which appeared to be eliminated as ECVs.” on page 4, 3rd paragraph.

On Pg. 10 and 12, the authors write:

“The second possibility is that the abnormal phenotype of CCRK-KO cells may be attributable to the lack of phosphorylation of CCRK substrates other than ICK, including MAK, although MAK is expressed in limited tissues [31, 79].” and “The candidate substrates include MAK, a tissue-specific paralog of ICK [31], and MOK, a more divergent paralog that is also expressed in a tissue-specific manner [79], both of which have a T(D/E)Y sequence. However, in view of the fact that Mak null mice are viable [93], it is likely that ICK, which is ubiquitously expressed [79], generally contributes to the regulation of ciliary functions."

Since MAK does not appear to be expressed in RPE cells and Mak mouse mutants are viable with the only known phenotype being in photoreceptor cells, it is probably not a good idea to mention MAK as an alternative potential mediator of CCRK’s functions (so, I’d remove it). In addition, it is not clear whether MOK expressed in RPE cells. If not, I would avoid mentioning MOK as a candidate mediator as well. Please be aware that CCRK substrates/mediators don’t necessarily have to be members of the ICK/MAK/MOK family.

According to these comments, we have removed the statements containing MAK and MOK, and revised the text before and after them to make sense. (page 10, the last paragraph; and page 12, 3rd paragraph)

Considering this second possibility put forward by the authors, they should consider that different ciliary functions of CCRK could be mediated by more than one factor and that the roles of CCRK mediators may vary depending on cellular context. In human RPE cells, it could be argued that many or all of CCRK’s functions are mediated by (still unidentified) factors other than ICK, given that forced ICK-T157E expression does not rescue the IFT88 tip accumulation phenotype of Ccrk mutants. However, in mouse (NIH3T3) fibroblasts, it appears that CCRK controls ciliary length simply by phosphorylating/activating ICK and toggling of ICK between phospho states is not important. This is supported by the fact that forced expression of ICK-T157E, but not wild-type or T157A ICK, is able to rescue the ciliary length phenotype of Ccrk knockdown NIH3T3 cells (Yang et al, EMBO reports 2013).

According to this comment, we have added “although a previous study using mouse NIH3T3 cells suggested that CCRK controls ciliary length simply by phosphorylating/activating ICK [56].” in Discussion. (page 12, 2nd paragraph)

---

## [Editor Report · Decision Letter 2]

29 Sep 2021

CCRK/CDK20 regulates ciliary retrograde protein trafficking via interacting with BROMI/TBC1D32

PONE-D-21-18831R2

Dear Dr. Nakayama,

We’re pleased to inform you that your manuscript has been judged scientifically suitable for publication and will be formally accepted for publication once it meets all outstanding technical requirements.

Kind regards,

Oliver E Blacque

Academic Editor

PLOS ONE

Additional Editor Comments (optional):

The authors have addressed the remaining comments
---

## [Editor Report · Acceptance letter]

1 Oct 2021

PONE-D-21-18831R2 

CCRK/CDK20 regulates ciliary retrograde protein trafficking via interacting with BROMI/TBC1D32 

Dear Dr. Nakayama:

I'm pleased to inform you that your manuscript has been deemed suitable for publication in PLOS ONE. Congratulations! Your manuscript is now with our production department. 

Kind regards, 

on behalf of

Dr. Oliver E Blacque 

Academic Editor

PLOS ONE